# Hypothalamic transcriptomes of 99 mouse strains reveal trans eQTL hotspots, splicing QTLs and novel non-coding genes

Yehudit Hasin-Brumshtein[1,2,3,4]*, Arshad H Khan[5], Farhad Hormozdiari[6], Calvin Pan[1,2], Brian W Parks[1,2,3,4], Vladislav A Petyuk[7], Paul D Piehowski[7], Anneke Brümmer[8], Matteo Pellegrini[9], Xinshu Xiao[8], Eleazar Eskin[6], Richard D Smith[7], Aldons J Lusis[1,2,3,4], Desmond J Smith[5]

[1]Department of Human Genetics, University of California, Los Angeles, Los Angeles, United States; [2]David Geffen School of Medicine, University of California, Los Angeles, Los Angeles, United States; [3]Department of Microbiology, University of California, Los Angeles, Los Angeles, United states; [4]Department of Immunology and Molecular Genetics, University of California, Los Angeles, Los Angeles, United States; [5]Department of Molecular and Medical Pharmacology, University of California, Los Angeles, Los Angeles, United States; [6]Department of Computer Science, University of California, Los Angeles, Los Angeles, United States; [7]Biological Sciences Division, Pacific Northwest National Laboratory, Richland, United States; [8]Department of Integrative Biology and Physiology, University of California, Los Angeles, Los Angeles, United States; [9]Department of Molecular, Cell, and Developmental Biology, University of California, Los Angeles, Los Angeles, United States

*For correspondence: yehudit.
hasin@gmail.com

**Competing interests:** The authors declare that no competing interests exist.

**Abstract** Previous studies had shown that the integration of genome wide expression profiles, in metabolic tissues, with genetic and phenotypic variance, provided valuable insight into the underlying molecular mechanisms. We used RNA-Seq to characterize hypothalamic transcriptome in 99 inbred strains of mice from the Hybrid Mouse Diversity Panel (HMDP), a reference resource population for cardiovascular and metabolic traits. We report numerous novel transcripts supported by proteomic analyses, as well as novel non coding RNAs. High resolution genetic mapping of transcript levels in HMDP, reveals both *local* and *trans* expression Quantitative Trait Loci (eQTLs) demonstrating 2 *trans* eQTL 'hotspots' associated with expression of hundreds of genes. We also report thousands of alternative splicing events regulated by genetic variants. Finally, comparison with about 150 metabolic and cardiovascular traits revealed many highly significant associations. Our data provide a rich resource for understanding the many physiologic functions mediated by the hypothalamus and their genetic regulation.

## Introduction

The regulation of body weight and appetite are complex processes, in which hypothalamic nuclei play a pivotal role. Genome wide association studies have shown that DNA sequence variants significantly contribute to variation in metabolic traits both in humans and mice. However, in most cases the connection between genetic variant and final phenotype remains unknown (*Suhre et al., 2011*; *Teslovich et al., 2010*; *Lappalainen et al., 2013*). In an effort to better understand how genetic variation results in phenotypic differences, many projects in the last decade have focused on genome

**eLife digest** Metabolism is a term that describes all the chemical reactions that are involved in keeping a living organism alive. Diseases related to metabolism – such as obesity, heart disease and diabetes – are a major health problem in the Western world. The causes of these diseases are complex and include both environmental factors, such as diet and exercise, and genetics. Indeed, many genetic variants that contribute to obesity have been uncovered in both humans and mice. However, it is only dimly understood how these genetic variants affect the underlying networks of interacting genes that cause metabolic disorders.

Measuring gene activity or expression, and tracing how genetic instructions are carried from DNA into RNA and proteins, can reliably identify groups of genes that correlate with metabolic traits in specific organs. This strategy was successfully used in previous studies to reveal new information about abnormalities linked to obesity in specific tissues such as the liver and fat tissues. It was also shown that this approach might suggest new molecules that could be targeted to treat metabolic disorders.

A brain region called the hypothalamus is key to the control of metabolism, including feeding behavior and obesity. Hasin-Brumshtein et al. set out to explore gene expression in the hypothalamus of 99 different strains of mice, in the hope that the data will help identify new connections between gene expression and metabolism.

This approach showed that thousands of new and known genes are expressed in the mouse hypothalamus, some of which coded for proteins, and some of which did not. Hasin-Brumshtein et al. uncovered two genetic variants that controlled the expression of hundreds of other genes. Further analysis then revealed thousands of genetic variants that regulated the expression of, and type of RNA (so-called "spliceforms") produced from neighboring genes. Also, the expression of many individual genes showed significant similarities with about 150 metabolic measurements that had been evaluated previously in the mice.

This new dataset is a unique resource that can be coupled with different approaches to test existing ideas and develop new ones about the role of particular genes or genetic mechanisms in obesity. Future studies will likely focus on new genes that show strong associations with attributes that are relevant to metabolic disorders, such as insulin levels, weight and fat mass.

wide characterization of sequence variants regulating gene expression in different tissues and organisms (*Lappalainen et al., 2013*; *Majewski and Pastinen, 2011*; *Grundberg et al., 2011*, *, 2012*). These studies showed that up to 80% of genetic variants associated with complex traits likely act through the regulation of gene expression rather than changing protein sequence and function. Such genes, termed expression quantitative trait loci (eQTLs), offer useful insights into the mechanistic links between genotype and phenotype, providing the eQTLs are characterized with sufficient power and resolution in phenotypically relevant tissues and states (*Pickrell et al., 2010*; *Min et al., 2012*; *Gaulton et al., 2015*).

Mouse is the primary model organism for many cardiovascular traits, including atherosclerosis, metabolic syndrome, obesity, the neural control of metabolism, and diabetes (*Lusis, 2000*; *Billon et al., 2010*; *Allayee et al., 2003*). Dozens of loci contribute genetically to metabolic traits have been identified in mouse models (*Parks et al., 2013*; *Lusis, 2012*). Indeed, we and others have extensively characterized eQTLs in metabolically relevant tissues of mice, suggesting potential new genes related to obesity (*van Nas et al., 2010*; *Keane et al., 2011*; *Huang et al., 2009*; *Aylor et al., 2011*). Integration of transcriptomic data from liver and adipose with genetic mapping and phenotypes led to mechanistic insights into the complexity of metabolic phenotypes. Yet, hypothalamus, which is not a readily accessible tissue, lacks such high resolution expression data. In fact, only one previous study examined eQTLs in mouse hypothalamus, using mice from 39 inbred strains fed chow diet, using microarray data and 5000 SNPs (*McClurg et al., 2007*). The lack of extensive transcriptome data, which allows mapping of eQTLs and the linking of traits and transcripts, is a major impediment in integrating the hypothalamus with a systems biology of metabolism. In this study, we used RNA-Seq to characterize the hypothalamic transcriptome in 99 inbred strains of mice

from the Hybrid Mouse Diversity Panel (HMDP, [*Bennett et al., 2010*; *Ghazalpour et al., 2012*]) fed a high fat, high sugar diet (*Parks et al., 2013*). The advantage of the HMDP is that these strains have all been densely genotyped and carefully phenotyped for about 150 metabolic traits, allowing high resolution genetic mapping of QTLs and eQTLs. We identified thousands of novel isoforms and hundreds of new genes that were not previously annotated, and that may represent variants or transcripts specific to the hypothalamus. The HMDP also allowed us to map QTLs with high resolution and power, identifying both *local* and *trans* acting variants. The RNA sequencing data permitted examination of a much broader spectrum of transcriptional features and facilitated analysis not only at the gene level, but also of genetic variants affecting specific isoforms, coding sequences or transcription start sites.

## Results

In this study, we explored the transcriptional landscape of mouse hypothalamus using RNAseq from 282 mice, representing 99 inbred and recombinant inbred strains from the HMDP. We first focused on quantifying gene expression in a transcript specific manner, the discovery of novel expressed regions and isoforms, and the contribution of genetic factors to expression variance. We also sought qualitative support for translation of new isoforms and genes. We then examined and quantified RNA modifications, such as alternative splicing events and editing in our data and used the data to map genetic variants affecting these events. Finally, we used the extensive phenotyping available for the HMDP, to look for associations between the expression of genes and transcripts, and physiologic phenotypes. All of the expression data and transcriptome assembly are publicly available from the Gene Expression Omnibus database, accession number GSE79551.

### Hypothalamic gene expression and proteomic data reveal multiple new isoforms and novel genes

Similar to other large-scale RNAseq studies (*Mutz et al., 2013*) we identified thousands of novel transcripts, with the vast majority of them only expressed at low levels in a small subset of samples (*Table 1*). Nevertheless, in the robust set of transcripts that are expressed at appreciable levels (FPKM >1 in 50+ samples and FPKM>0 in 100+ samples), we still identified 21,234 novel isoforms and 485 transcripts originating from 407 novel expressed genes (*Supplementary file 1*).

**Table 1.** Transcriptome assembly and filtering.

|  | None | Filter #1* | Filter #2[†] | NR[‡] |
|---|---|---|---|---|
| Loci (genes) | 40472 | 37591 | 14079 | 14079 |
| Transcripts | 383420 | 357066 | 51024 | 50347 |
| known (=) coding | 99658 |  | 20721 |  |
| known (=) non coding |  |  | 8584 |  |
| novel isoform (j) | 259570 |  | 21234 |  |
| novel locus (u) | 11753 |  | 485 |  |
| other status | 12439 |  |  |  |
| TSS | 100073 | 94417 | 32537 | 20013 |
| CDS | 46242 | 46242 | 18687 | 7643 |
| Total features | 570207 | 535316 | 116327 | 92082 |

*Filter #1: Expression values of <1e–6 were rounded to zero, and *novel* transcripts with all zero values were removed both from expression table and from merged file. Also, all transcripts with class code not "=", "j" or "u" were removed.

[†]Filter #2: Implementation of detection and expression thresholds (detected in more than 100 samples and expressed (fpkm>1) in more than 50) on each feature separately.

[‡]Filter_NR: Non redundant features count (those that do not have 1_2_1 correlation to a gene).

Interestingly, the number of novel isoforms in our study is comparable to the number of previously annotated transcripts passing the same filtering criteria (n = 29,305, *Table 1*).

There are several possible interpretations to as why the 407 genes could be missing from the genomic annotation. First, the 407 novel genes expressed in our data potentially constitute transcripts that are unique to the hypothalamus. Second, the GENCODE M2 annotation used in this study was the most recent available when we started to analyze the data. Since then, however, the annotation has been augmented based on more recently published datasets and improved prediction pipelines. Thus, while the 407 genes are novel relative to the M2 version, they may have been added later. To explore that possibility, we compared our assembly to the latest version of annotation released by GENCODE – M10 (released January 2016, http://www.gencodegenes.org/) and find that 193 out of 407 loci are still novel. Some of the genes may also represent genomic DNA contamination. However, we consider this possibility less likely since we used stringent filtering criteria based on the number of samples that the genes were expressed in.

In terms of genomic properties, such as putative open reading frame length, transcript length or splicing complexity, the novel genes seem to resemble known non-coding genes, suggesting that the majority of the novel genes likely belong to this class. On the other hand, the properties of novel isoforms are consistent with known coding transcripts (*Figure 1*). To explore the translational potential of newly identified isoforms and genes, we compared our RNA-Seq data to proteomic data generated from additional hypothalamus samples from the HMDP. Specifically, we translated all known and newly identified transcripts in 6 potential open reading frames, and compared this dataset to the hypothalamic peptide sequences. As expected, >95% of the identified peptides (*Supplementary file 1*) matched at least one known transcript with the vast majority of

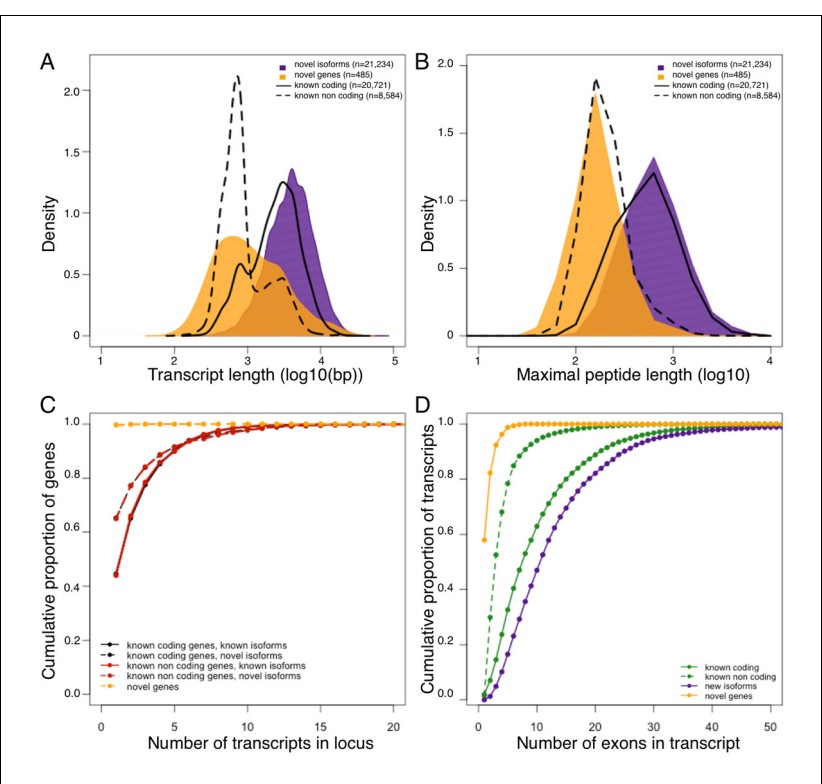

**Figure 1.** Genomic properties of novel genes are similar to known non coding genes. Novel genes and isoforms are defined by Cufflinks class code 'u' and 'j' respectively. Distributions of transcript length (**A**), and maximal hypothetical peptide length (**B**) of novel genes (yellow), new isoforms (purple), known non coding transcripts (dashed line) and known coding transcripts (solid line). Transcriptional complexity (number of transcripts per locus, (**C**) and splicing complexity (number of exons per transcript, (**D**) of novel genes, novel isoforms, known coding and know non coding transcripts.

these (>99%) corresponding to known protein coding transcripts, suggesting high accuracy of the peptide data. In addition, 430 peptides matched either novel isoforms (n = 401), or novel genes (n = 29) exclusively (*Table 2*). Since the genomic properties of the novel genes hint that the majority of them are likely non coding, we do not find their low representation in peptide data surprising. Moreover, manual in-depth characterization of the 29 peptides matching novel genes, with UCSC and NCBI database revealed that almost all of them represent processed pseudogenes, rather than proteins with novel functions. This finding both supports the previously published reports of wide spread transcription of pseudogenes, and their translation (*Kim et al., 2014*; *Tay et al., 2015*), while strengthening the suggestion that the majority of novel genes identified in this work are not protein coding.

We further examined whether multi-mapping reads are a substantial contributor to the measured expression of the novel identified genes. For that purpose, we re-quantified 3 samples from C57BL/6J strain, using uniquely mapping reads only (mapping quality = 255), and compared the quantification of the novel genes between the two approaches. Not surprisingly, the proportion of uniquely mapped reads contributing to the expression of the genes matched very well between the 3 samples, suggesting that it is an intrinsic gene property and not unique to the individual samples. Importantly, while the low uniqueness of mapping reads may indicate false results, we also noted that among the 29 peptides matching to the novel genes exclusively and uniquely, 18 match genes that do not pass our threshold. In fact, many of gene families share sequence motifs and are homologues, and as such some of the reads would be multi-mapping between them. Thus we chose not to remove these genes from the annotation.

## Genetic regulation of gene expression

We examined expression QTLs in terms of type of the affected features and identified SNPs affecting expression of all transcripts at a locus (eQTL), specific transcript isoforms (isoQTLs), transcription start sites (tssQTLs) or open reading frame (cdsQTLs) (*Hasin-Brumshtein et al., 2016*). Since the linkage disequilibrium (LD) is extensive in the mouse genome, we distinguish between three types of QTLs: *local* (within 2 Mb of the gene), *distant* (on the same chromosome as the gene, distance >2 Mb) and *trans* (SNP and gene reside on different chromosomes). The number of identified interactions, and genes affected by such regulation depends on the statistical cutoff of p-value for the interaction one chooses. We examined a set of thresholds ranging from very liberal to stringent. The liberal threshold was previously established by permutation from microarray expression data in the HMDP, i.e. p<1.4E–3 for *local* variants and p<6E–6 for *distant* and *trans*. The most stringent threshold used a Bonferroni corrected threshold (i.e 0.01 divided by the number of tests, p<1E-12). As expected, different thresholds resulted in different numbers of QTLs, with *local* and *distant* QTLs being more robust to threshold stringency than *trans* QTLs. However, 85–95% of the genes regulated by *distant* variants, were also regulated by *local* variants, suggesting that most of the *distant* QTLs reflect *local* signals emulated at a distance as a result of wide-ranging LD, rather than independent regulatory elements. In contrast, 70% of the *trans* interactions involved genes lacking *local*

**Table 2.** Summary of peptide support for transcripts.

| Peptide matching* | All | Uniquely mapped |
|---|---|---|
| Known transcripts | 9913 | 1016 |
| Protein_coding | 9839 | 1002 |
| ncRNAs | 74 | 14 |
| Novel isoforms | 401 | 94 |
| Novel genes | 29 | 24 |
| Total number | 10343 | 1134 |

*Peptides are counted towards supporting novel isoforms only if they do not match any known transcript, and towards supporting novel genes if they do not match either known or novel isoform transcripts. All peptides matching known transcripts were also assigned the most likely transcript. Non coding status was assigned only to peptides that do not match any coding transcripts (for full details please see Materials and methods).

signal over the entire set of thresholds. Since *distant* regulation was mostly redundant to *local*, and it would be very difficult to determine whether that signal is a result of independent regulation versus extended LD, we chose to focus our analysis on *trans* and *local* interactions only, defining *trans* as trans-chromosomal interactions. If we look at regulation types over a wide range of thresholds, the *local* signal predominates at the more stringent thresholds reflecting the larger typical effect size in this group (*Figure 2A*). We identified *local* and *trans* QTLs for all expressed feature types, with the most common being eQTLs and isoQTLs (*Figure 2B*). Notably, while we used kinship matrix specific for our strains, still several genes in our dataset show extensive *trans* regulation (horizontal lines in *Figure 2B*) suggesting a residual influence of population structure on our mapping results. We also note that regulation of gene expression often occurs at the gene level, than at transcript specific levels (*Figure 2C*).

Classical genome-wide eQTL studies used association analysis of total gene expression levels to map *local* eQTL, assuming that variation linked to proximal genetic variant indicates *cis* regulation. Recently, several studies have exploited the single base resolution of RNA-Seq to examine this assumption. RNA-Seq permits the identification allele expression (ASE), a hallmark of functional *cis* acting regulation, in heterozygotes, such as humans or in mouse F1s. Notably, these studies generally show poor concordance between ASE ratios and previously identified *local* eQTLs, which had

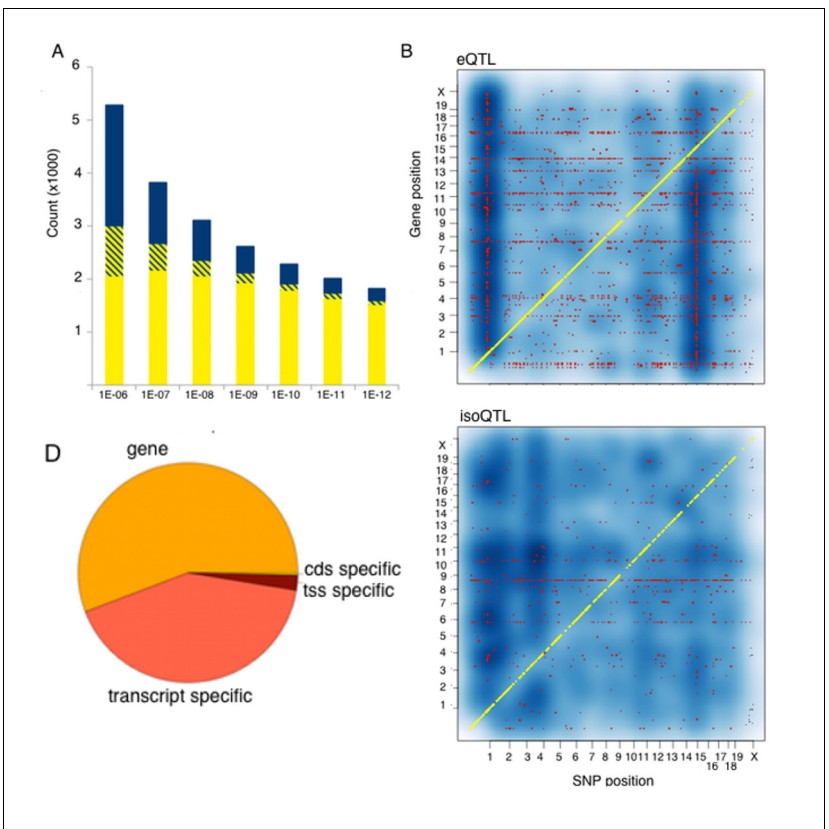

**Figure 2.** Genetic regulation of expression in the mouse hypothalamus. (**A**) Number of genes affected by *trans* (blue), *local* (yellow) or both (striped) variants as a function of statistical threshold. (**B**) Gene level expression quantitative trait loci (eQTL, top) but not transcript specific (isoQTL, bottom) show *trans* eQTL hotspots. Density shows the number of interactions at lower statistical thresholds (1e–6), red shows interactions passing 1E-12 threshold. Yellow indicates cis acting variants. (**C**). Genetic regulation occurs on every level, but gene level regulation is more prevalent than transcript specific cases. Supplementary figure shows correlations between allele expression in F1 and local eQTLs identified in HMDP hypothalamus.

The following figure supplement is available for figure 2:

**Figure supplement 1.** Allele specific expression in whole brain correlation to local eQTL.

been attributed mostly to different technical aspects of RNAseq and ASE analysis (*Hasin-Brumshtein et al., 2014*; *Lappalainen et al., 2013*; *Grundberg et al., 2012*). To examine the concordance between *local* eQTLs obtained from ASE and genome-wide association, we performed RNA-Seq on brain tissue from 20 mice representing two F1 crosses between three of the classical inbred strains used to construct the HMDP. The parental strains were C57Bl/6 (B), A/J (A) and C3H (H), and the F1 offspring were BxA and HxB. We compared the effect size of ~2600 *local* eQTLs, to the average ASE effect in both crosses. While *local* eQTLs were significantly positively correlated with ASE (p<2E-16) correlation estimates were modest ($R^2$ = 0.2), and lower than between the ASEs in the two sets of F1 mice ($R^2$ = 0.4, *Figure 2—figure supplement 1*).

## RNA-Seq shows extensive alternative splicing in the hypothalamus, with complex regulation pattern

Using a previously established pipeline for analysis of alternative splicing events, SUPPA (see Materials and methods), we identified 7564 alternative splicing events affecting 3599 genes (*Table 3*). An alternative first exon was most common, accounting for the majority of alternative splicing events with several genes exhibiting multiple alternative first exons (*Figure 3A*). For other types of events, each most often affected one gene, with some of the genes exhibiting a combination of alternative splicing events. The extent of alternative splicing in each sample was quantified as a percent spliced in (PSI) of every event (*Figure 3B*). This quantification can be regarded as a quantitative trait; however, the distribution of PSI for every type of event suggests an excess of 0 or 1 values (never or always spliced in, respectively *Figure 3E*). This observation is consistent with alternative splicing being often bimodal rather than a normally distributed continuous measure (e.g. present or absent splice site), however, it also shows that quantitative regulation plays a significant role in ratio of isoforms that arise from the inclusion or exclusion of a splicing event.

Contrary to gene expression, all forms of alternative splicing were predominantly regulated by *local* variants (*Figure 3C,D*). This is not unexpected, since variation in alternative splicing is most likely to arise from particular sequence variants in the RNA itself. Across all examined categories of alternative splicing, 50–60% of the events were significantly associated with *local* variants, the strongest signal often residing within 1 Mb of the event (*Figure 3F*). We observe that for many of the genetically regulated splicing events the data show an excess of 0 or 1 PSI values that correlate with the genetic variant (e.g *Figure 3G*), rather than a shift in a continuous quantitative distribution. This observation suggests that many of the genetically regulated events in alternative splicing sequence play a deterministic role.

## Trans eQTL hotspots

*Trans* eQTLs are not uniformly distributed across the genome, clustering into potential hotspots of genome wide regulation. Such hotspots have been observed in several datasets (*Orozco et al., 2015*; *Orozco et al., 2012*; *Tian et al., 2015*), and they are thought to represent cases where a *cis* acting variant affects a regulatory gene, e.g. transcription factor, subsequently affecting expression

**Table 3.** Alternative splicing events.

|  | All events | Cis QTL events |
|---|---|---|
| Alternative 3' splice site (A3) | 316 (266) | 155 (134) |
| Alternative 5' splice site (A5) | 365 (304) | 220 (189) |
| Alternative first exon (AF) | 5288 (1874) | 2776 (1278) |
| Alternative last exon (AL) | 507 (320) | 283 (208) |
| Mutually exclusive exons (MX) | 44 (36) | 29 (27) |
| Retained Intron (RI) | 645 (476) | 356 (285) |
| Skipped exon (SE) | 399 (323) | 221 (189) |
| Total | 7564 (3599) | 4040 (2310) |

*Table 2*: Number in parenthesis indicates number of distinct genes affected by the events.

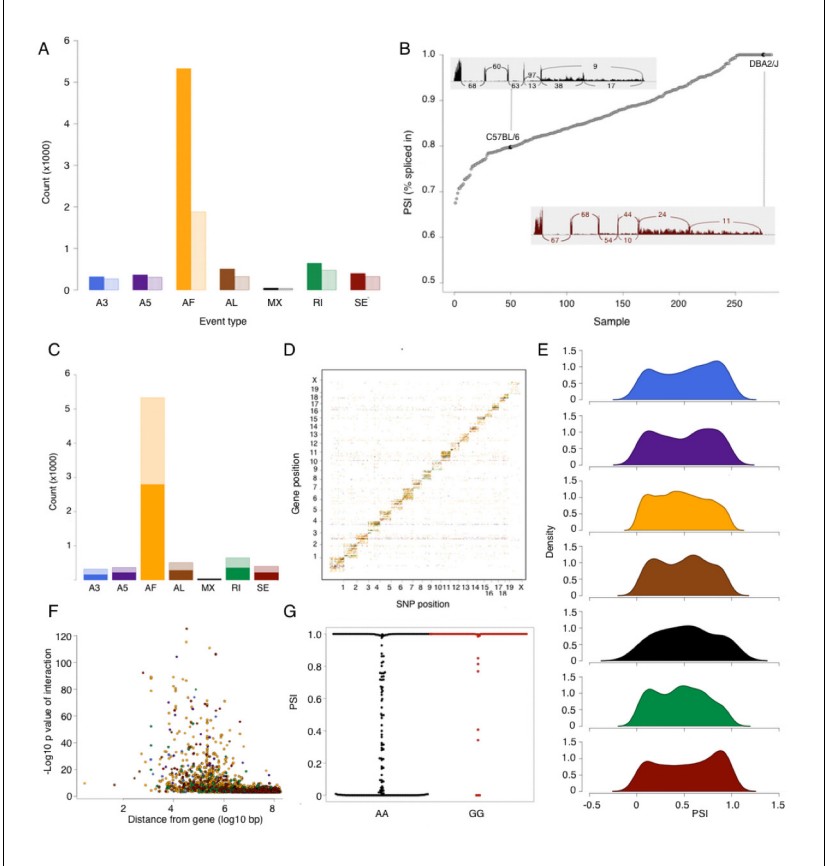

**Figure 3.** Alternative splicing in the mouse hypothalamus. Alternative splicing events were classified (see Materials and methods) into 7 types: alternative 3' splice (A3, blue), alternative 5' splice (A5, purple), alternative first exon (AF, orange), alternative last exon (AL, brown), mutually exclusive exons (MX, black), retained intron (RI, green), and skipped exon (SE, dark red). All events were quantified in each sample for percent spliced in (PSI). (**A**) Number of alternative splicing events of each type (solid color), and number of genes affected by these events (light color). (**B**) Example of partial exon skipping in *Colq* gene. DBA shows the complete inclusion of the exon (therefore PSI = 1), while in C57BL/6 there is partial exon skipping (PSI = 0.78). (**C**) Number of alternative splicing events with and without *local* QTL signal (solid and light color respectively). (**D**) Alternative splice QTLs are mapping to the same chromosome, for all types of events, indicating that most of genetic regulation is by *local* (and likely *cis* acting) variants. (**F**) Distance between most significant SNP for each event and gene start. The largest effect is typically within 1 Mb of the gene. (**G**) An example of mapping of mutually exclusive exon event in *Nnat* gene mapping to SNP rs32019082. (**E**) Distribution of all PSI values of each event type in all samples.

of multiple targets. To identify *trans* hotspots we divided the genome into bins of 100 kb, and counted the number of distinct genes which exhibit a *trans* interaction associated with any of the SNPs in the bin (*Figure 4A*). Importantly, the bins do not necessarily contain the same number of SNPs, and are an order of magnitude shorter than the typical LD blocks in the inbred mouse genome. Just counting the number of SNPs in a bin that are associated in *trans* would be confounded by SNP density. However, since LD essentially recapitulates the same interactions over and over again, counting the number of genes per bin, rather than the number of SNPs, is not significantly affected by LD.

There are clearly two very strong *trans* acting hotspots on chromosome 1 and 15, which are observable for eQTLs, but not for isoQTLs, tssQTLs or cdsQTLs. Each of the two hotspots span 5–6 Mb and include >500 SNPs regulating >400 genes in *trans* (*Figure 4B*, *Supplementary file 2*). Functional enrichment analysis of gene targets of these hotspots suggests that the hotspot on chr1 regulates multiple genes involved in nucleotide binding, while genes regulated by chr15 locus are clearly associated with ion transport in synapse activity (*Figure 4C*, *Supplementary file 2*). Moreover,

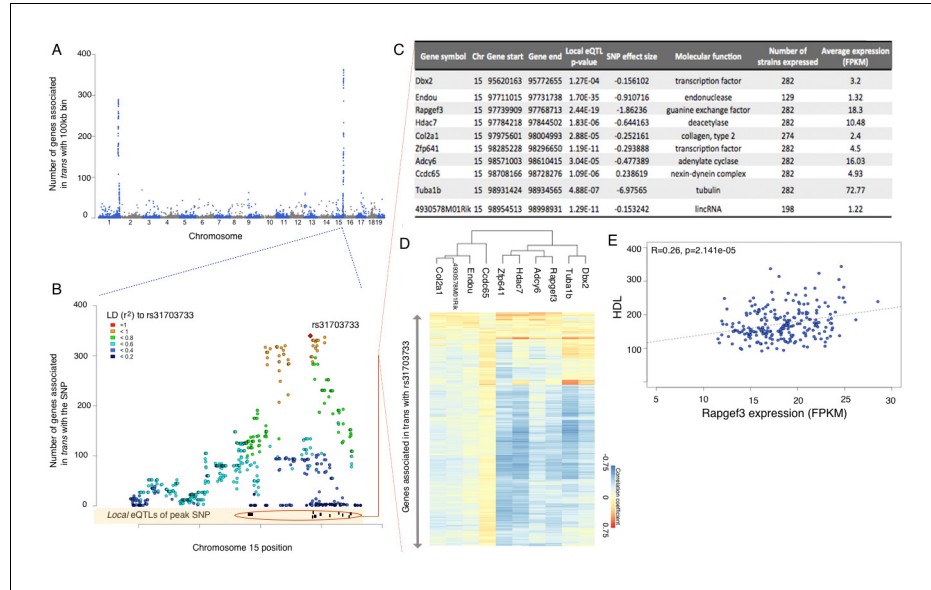

**Figure 4.** eQTL mapping suggests trans eQTL hotspots in the hypothalamus regulate expression of hundreds of genes. (**A**) Mouse genome was broken into 100 kb bins. The plot presents genome wide counts of genes which expression is associated with SNPs in that region, in *trans*. (**B**) Zoom of trans eQTL locus on chromosome 15. Peak SNP (associated with most genes in *trans*, rs31703733) is shown in red, color of other SNPs indicates $r^2$ to rs31703733. Lower track shows the 10 genes which expression is associated with rs31703733 *locally*. C,D,E pertain to the 10 genes associated *locally* with rs31703733 and therefore potentially mediate the *trans* effects. (**C**) Summary table about each gene. (**D**) Heatmap showing correlation of expression between genes associated with rs31703733 locally, and genes associated with rs31703733 in *trans*. Color indicates Pearson correlation coefficient. (**E**) Example correlation between potential regulator (RApgef3) and trait (HDL levels).

consistent with the hypothesis that *trans* regulation is mediated by local effects, the *trans* acting SNPs in these hotspots, are also associated with expression of several local genes (*Figure 4B,C*, *Supplementary file 2*).

To identify potential mediators of *trans* regulation, we examined the *trans* eQTL hotspot on chr15 in more detail. In this region a cluster of SNPs, which share high linkage disequilibrium, is associated with the expression of the majority of the genes mapping to this locus in *trans* (*Figure 4B*). We focused on the SNP with most *trans* associations (rs31703733), and found that while the hotspot itself contains dozens of genes, rs31703733 constitutes a *local* eQTL for only 10 of them (*Figure 4B*). Moreover, the expression levels of genes regulated in *trans* by the hotspot as well as 6 genes that were *local* eQTLs were closely correlated (*Figure 4C*), forming a coexpression module. Further, this module was significantly associated with triglycerides and cholesterol measurements in the mice. Interestingly, the two strongest *local* eQTL genes for this hotspot, *Endou* (an endonuclease) and *Rapgef3* (also known as *EPAC*, cAMP activated guanine exchange factor involved in *Ras* signaling) were significantly associated with cholesterol and triglyceride measures in this study as well.

Genetic association does not necessarily imply that the associated allele is causative for the change in expression. In fact, the majority of the eQTL SNPs fall into regions with poor or no annotation, making it unlikely to be the actual causative variant. However, surprisingly, peak SNPs in the chr15 hotspot (SNPs rs31703733 and rs31780670) are located within a H3K4me1 histone methylation mark associated with enhancers, and which was detected in all neuronal tissues probed by ENCODE in adult mice (cortex, cerebellum, olfactory bulb), but not in other metabolic tissues such as liver, heart, intestine or lung (UCSC genome browser). Based on this preliminary and indirect evidence one may hypothesize that rs31703733 and rs31780670 potentially affect the enhancer activity and the expression of nearby genes. However, to reach any functional conclusion based on our data would require experimental validation in model systems.

## Long non-coding RNAs in the mouse hypothalamus

Long non coding RNA (lncRNAs) are a generally poorly characterized class of RNA molecules, with sometimes unclear classification (*St Laurent et al., 2015*). Commonly used criteria for identification of lncRNAs are a transcript >200 bp long without an obvious potential open reading frame. In contrast to the relative paucity of data regarding the functionality of most lncRNAs, several specific lncRNAs (e.g. Xist, HOTAIR or H19) had been studied extensively and shown to play an essential role in cellular function (*Quek et al., 2015*). Recent large RNA-Seq studies and integrative projects such as ENCODE suggest that lncRNAs likely constitute up to 60% of the transcribed RNAs (*Djebali et al., 2012*). Moreover, in recent years an increasing number of functional studies have shown that lncRNAs play an important role in the regulation of transcription and translation, as well as in cell differentiation, signaling and other processes (*Sun et al., 2013*; *Guttman et al., 2011*; *Ramos et al., 2015*). Further, lncRNAs are enriched for genetic association signals in genome wide association studies (*Iyer et al., 2015*; *Kumar et al., 2013*).

GENCODE M2 annotation classifies 4540 lncRNA transcripts into at least 5 biotypes, based on their overlap with protein coding genes (http://www.gencodegenes.org/mouse_releases/2.html). The two most common biotypes - long intergenic non coding RNAs (lincRNAs) and antisense RNAs, account for 97% of all the annotated lncRNAs. Since our data were generated without retaining strand specificity, we focused on lincRNAs only. LincRNAs have been shown to be highly tissue specific (*Cabili et al., 2011*), and therefore it is not surprising that although there are 2417 annotated lincRNA transcripts, we found only 381 expressed in the mouse hypothalamus at our filtering criteria. The 381 transcripts represent 237 known and 144 novel isoforms of 181 lincRNA genes, with both novel and known isoforms showing similar expression levels (*Figure 5A,B*). We also used length and open reading frame criteria to examine the novel loci. Notably, the RNA length criteria of >200 bp is a commonly accepted parameter, while the length of minimal potential open reading frame varies between 30–100 aa in different studies. Based on these criteria we can identify up to 129 potentially novel lincRNAs expressed in the mouse hypothalamus, with 49 of them being spliced. Since lincRNAs have been shown to be often highly tissue specific, we consider the novel lincRNAs to be good candidates for hypothalamus specific transcripts.

We used gene expression levels to explore the correlation of 310 lincRNAs (181 known and 129 novel genes) with metabolic traits in HMDP. Up to 35% of lincRNA transcripts significantly (p<1e–3) correlated to at least one phenotype in the HMDP, with a few lincRNAs associated with a multitude of related phenotypes (*Figure 5C,D*). The statistical threshold was determined by permutations in

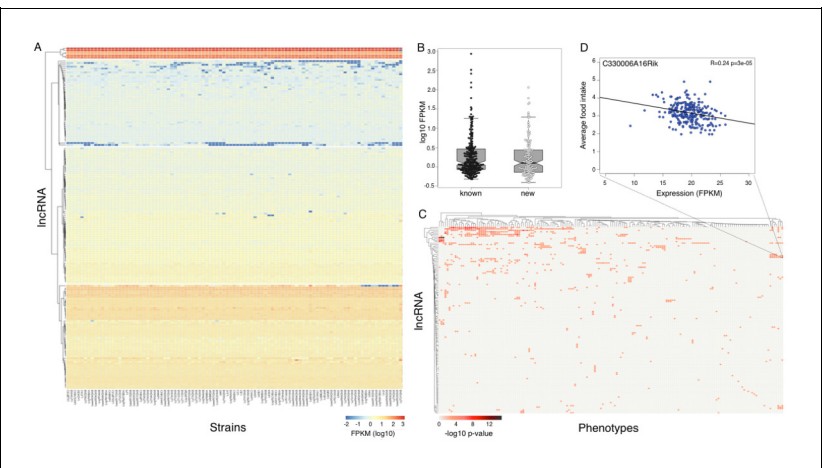

**Figure 5.** Expression of long non coding RNAs in the hypothalamus is phenotypically relevant. (**A**) Expression of heatmap known non coding RNAs and novel isoforms of these genes n the HMDP. Six lncRNAs (top cluster, Meg3, Gm26924, Snhg4, Miat, 6330403K07Rik, and Malat1) are highly expressed in almost all samples. (**B**) Novel isoforms of lncRNAs are expressed at a similar level of known ones. (**C**) Long non coding RNAs are associated with multiple phenotypes in the HMDP. (**D**) An example of association between a non coding RNA C330006A16Rik and average food intake.

previous studies of gene expression in HMDP, is less stringent than a Bonferroni correction, and reflects interdependencies among phenotypes. Expression of >30% of lincRNAs is associated with a significant *local* eQTL, suggesting that a considerable number of lincRNAs in the hypothalamus are playing an important role in translation of genetic regulatory variance into physiologic phenotypes. Notably, six lincRNAs – *Meg3*, *Gm26924*, *Snhg4*, *Miat*, *6330403K07Rik*, and *Malat1*– were highly expressed in most strains (*Figure 5A* upper cluster). *Meg3* is a known imprinted tumor suppressor gene (*Zhang et al., 2010*) and was recently implicated in hepatic insulin resistance (*Zhu et al., 2016*). *Miat* and *Malat1* are both part of nuclear bodies. Knockout models of *Malat1* are all viable and normal (*Zhang et al., 2012*; *Eißmann et al., 2012*; *Nakagawa et al., 2012*), however their response to metabolic challenges, such as high fat diet, had not been assessed. Myocardial infarction associated transcript (*Miat*) was initially linked to myocardial infarction through a genetic association study(*Ishii et al., 2006*). Subsequently, *Miat* was shown to regulate development of neuronal progenitors, involved in schizophrenia pathogenesis and fear response(*Aprea et al., 2013*; *Liao et al., 2016*). Although all of the six highly expressed lincRNAs are not novel, none of them were previously reported to be expressed in the hypothalamus, or to play an established role in the metabolic or reproductive system.

Previous investigations have documented several examples of lincRNAs that code for short translated open reading frames (*Anderson et al., 2015*). In addition, recent work has shown that lincRNAs can be associated with ribosomes, and their occupancy is similar to that of protein coding transcripts (*Ingolia et al., 2014*; *Ruiz-Orera et al., 2014*). We detected 6 peptides which mapped uniquely to 3 different lincRNAs (Gm26825, Gm16295, and Gm26593), suggesting that translation of lincRNAs occurs rarely, and that their association with ribosomes is more likely to be in the context of translational regulation of other protein coding transcripts.

## RNA editing in the mouse hypothalamus

RNAseq provides the opportunity to look at RNA sequence modifications in a quantitative manner. In this study, we examined RNA editing patterns of all possible substitution types over the genome. A recent study showed that genetic variation affects C to U editing in the intestine, both in site specific and genome wide manners in the mouse Diversity Outbred cross (*Gu et al., 2015*). We hypothesized that genetic variation may contribute to the extent of RNA editing either in a site specific (e.g. by altering editing sites in *cis*) or genome wide manner (e.g. by altering the expression or specificity of editing enzymes).

Altogether we detected 8462 potential editing sites in 3319 genes. As expected, the majority of edits (>70%) were A-to-G modifications, and the total number of detected sites in each strain varied between several hundred to over a thousand (*Figure 6A*). A comparison of our findings to previously reported editing sites in two databases - DARNED and RADAR - suggests that 75% of these are novel. We then used the proportion of edited reads per site, or per substitution type, across strains as a quantitative trait for mapping genetic variants that contribute to editing. We did not detect any significant genetic association for genome-wide levels of editing, which is consistent with both lack of genetic variation in ADAR in our panel and its very stable expression level across samples (49 ± 0.27 FPKM).

Most editing sites were detected only in a small number of strains, which precluded meaningful mapping. For the site specific mapping we therefore focused on 90 sites that were detected in >70% of the strains (*Figure 6C*). We detected editing QTLs for 3 specific sites. For example, one of the editing sites in *Ociad1* (on chr5 at 73312444) had a strong association (p<1e–8) with genetic variation residing on the same chromosome (*Figure 6D*). Altogether our data suggest that RNA editing occurs at low level in thousands of genes, however the impact of genetic variation on the editing level in mouse brain is limited.

## Association of hypothalamic expression and phenotypes

There are 150 phenotypes available for the HMDP, reflecting many clinically relevant traits as well as various metabolic measures (*Parks et al., 2013*). We used several approaches to examine the potential role of hypothalamic expression in the various phenotypes. We began by identifying the top 500 transcripts that were significantly correlated with each of the phenotypes (correlation p values ranged from 4e–3 to <2e–16, *Supplementary file 3*). We then analyzed each of these expression

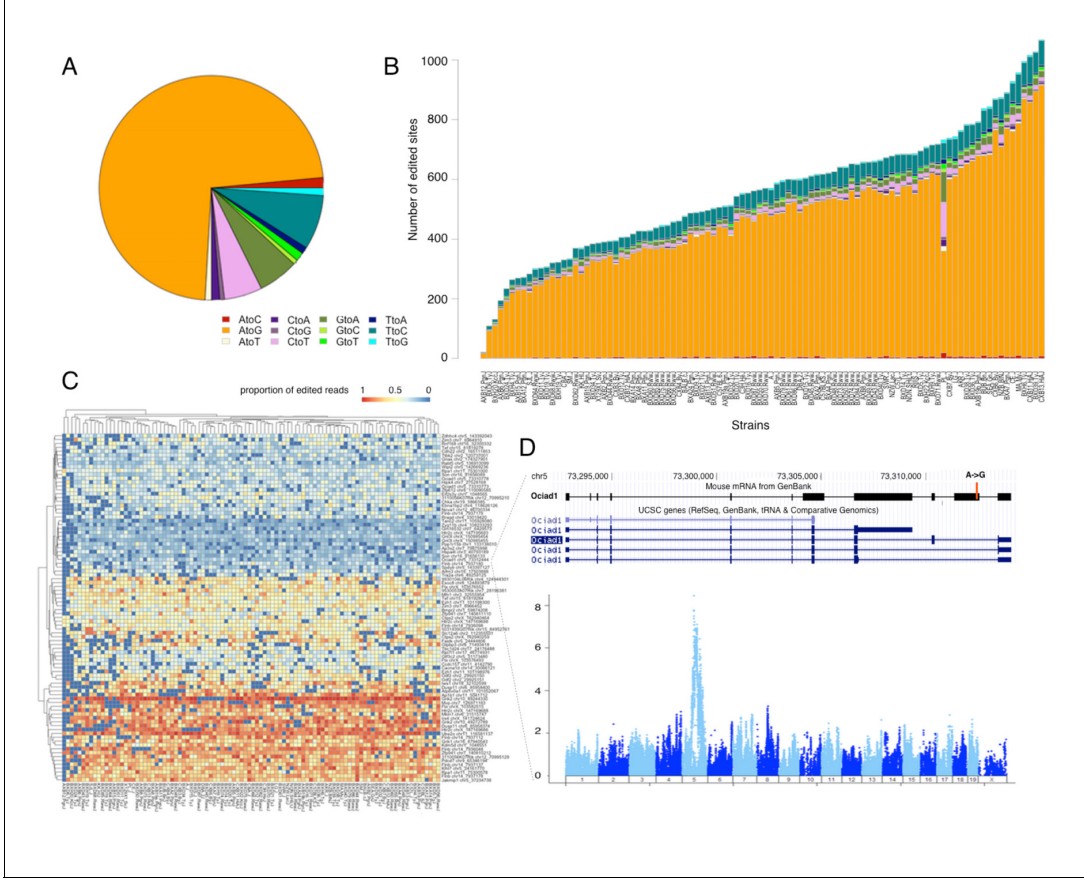

**Figure 6.** RNA editing is prevalent at the mouse hypothalamus at low levels. (**A**) A total of 8462 editing sites were identified in the HMDP, with A to G accounting for >70% of the modifications. (**B**) Number of sites identified in each strain (color coding as in **A**). (**C**) Editing level at 90 sites, that were detected in at least 70% of the samples, were mapped. Heatmap shows variation in editing in these sites among the strains. (**D**) An example of an edited site in Ociad1 gene, and its genome wide mapping result.

sets for enrichment of potential pathways and functional annotation, using DAVID (*Huang et al., 2009a*, *2009b*). Surprisingly, only a few of those expression sets showed moderate to strong enrichment (*Figure 7B*). For example, fat mass accumulation between 4 and 8 weeks, was distinctly associated with pathways related to oxidative phosphorylation and energy metabolism, while genes associated with food intake were enriched for ribosome related pathways. Clustering of phenotypes based on the proportion of shared genes associated with each trait (*Figure 7A*), clearly showed that related phenotypes also shared expression dependencies. For example, up to 70% of the genes most correlated with 'esterified cholesterol', were also correlated with 'total cholesterol', and more than 30% were shared with 'HDL'. Together these data validate the notion that related phenotypes share underlying molecular mechanisms, yet these shared genetic correlations may not necessarily correspond to specific readily identifiable pathways.

We then analyzed whether the novel genes and isoforms we uncovered potentially contribute to phenotypic diversity. We found 232 novel transcripts associated with at least one phenotype (*Supplementary file 3*). For example TCONS_00279690, was strongly associated with several metabolic traits, such as total mass in response to a high fat, high carbohydrate dietary challenge, initial mouse insulin levels, as well as weight and fat mass (*Supplementary file 3*). Similar to known isoforms, expression of ~30% of the novel isoforms was significantly correlated with phenotypes. Correlation coefficients and p values were also similar between known and novel isoforms, suggesting that novel isoforms are as likely to contribute to mouse diversity as previously identified transcripts.

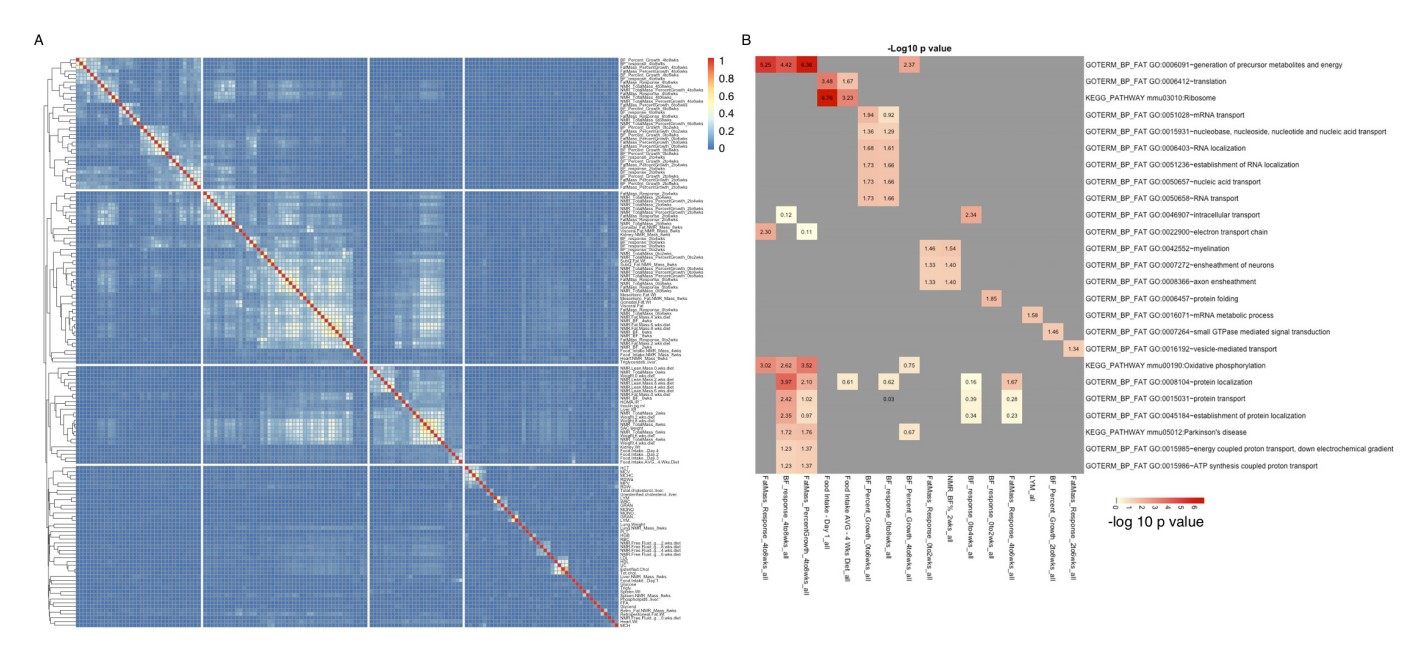

**Figure 7.** Groups of genes are associated with multiple related phenotypes in HMDP, although not necessarily enriched for GO ontology or specific pathways. (**A**) Fraction of co-shared genes among the 500 genes most associated with a phenotype. (**B**) Enrichment analysis of the top 500 genes associated with each of the 150 phenotypes results in a small number of significant associations.

## Expression of cell population markers in the hypothalamus

The hypothalamus is a heterogeneous brain region containing multiple nuclei that affect different aspects of metabolism and endocrine physiology. One of the drawbacks of our approach, dictated by practical considerations, is that we performed RNA-Seq on the entire hypothalamus rather than particular cell populations or nuclei. This approach likely results in dilution of signals from any one population of cells. Nevertheless, the sensitivity of RNA-seq allows examination of expression of particular cellular markers that are specific to certain cell types. To assess which cell populations are well represented in our data, we examined the expression, genetic regulation and association with phenotypes of some of the known markers of hypothalamic cell populations (*Table 4*). Our data show high expression of well-known hypothalamic neuronal markers, such as Agouti-related protein (Agrp), pro-opiomelanocortin (Pomc), hypocretin (orexin, Hcrt) and steroidogenic factor-1 (SF1). In addition, we detect high expression of oligodendrocyte markers and microglia, but only modest expression of epithelial markers.

Interestingly, we identified genetic variation affecting the expression of some key functional molecules for metabolic regulation and response to high fat diet. For example, myeloid differentiation primary response 88 factor (Myd88), is a Toll-like receptor (TLR) adaptor molecule. This protein mediates fatty acid induced inflammation and leads to leptin and insulin resistance in the central nervous system. Mice with CNS specific deletion of Myd88, are protected from high fat diet induced weight gain, and development of leptin resistance induced by acute central application of palmitate (*Kleinridders et al., 2009*). Our data show that there is a strong eQTL (p value 1.9e–17, *Table 4*) modulating expression of Myd88. In addition, we detect genetic variations that affect expression of key molecules such as leptin receptor (LepR) and Pomc. Together, these data suggest that the genetic background of inbred mice is an important factor in functional studies, and that the results of molecular perturbations of hypothalamic metabolic pathways can be modulated by genetics.

In addition to exploring the links between genetic variation and expression traits, we also looked into the association of transcript levels with phenotypes. We confirmed known relationships – for example the expression of orexigenic neuropeptide Y (NPY) was associated with total body mass (pvalue = 2.4e–6). Interestingly, one of the strongest correlations we observed were between

**Table 4.** Expression of hypothalamic markers.

| Gene | Marker for | Mean expression | local eQTL |
| --- | --- | --- | --- |
| AgRP | ARC neurons | 119.040 | ND |
| NPY | ARC neurons | 18.162 | ND |
| Foxo1 | ARC neurons | 4.736 | ND |
| POMC | ARC neurons | 271.954 | 2.618E-04 |
| Hcrt (orexin) | LHA neurons | 128.898 | 2.053E-05 |
| Sf1 | VMHvl neurons | 65.333 | ND |
| Nkx2-1 | VMHvl neurons | 6.067 | 1.100E-05 |
| Tac-1 | VMHvl neurons | 36.373 | 1.624E-09 |
| BDNF | VMHvl neurons | 4.464 | ND |
| Esr1 | Multiple | 2.658 | ND |
| LEPR | Multiple | 1.644 | 2.041E-04 |
| INSR | Multiple | 9.496 | ND |
| CX3CR1 | Microglia | 9.790 | 1.400E-05 |
| AIF-1 | Microglia | 98.598 | ND |
| CD68 | Microglia | 4.162 | ND |
| Itgam | Microglia | 3.599 | ND |
| MyD88 | Microglia | 2.369 | 1.942E-17 |
| Aqp4 | Astrocytes | 30.041 | ND |
| Slc1a3 | Astrocytes | 120.203 | ND |
| Aldh1l1 | Astrocytes | 16.787 | ND |
| Gfap | Astrocytes | 72.502 | 2.140E-06 |
| VEGF-A | Tanycytes | 39.559 | ND |
| CX3CL1 | Neurons | 99.411 | 9.836E-09 |
| Mog | Oligodendrocytes | 34.569 | 1.807E-04 |
| Mbp | Oligodendrocytes | 1266.574 | ND |
| Plp1 | Oligodendrocytes | 569.620 | ND |
| Car4 | Endothelial | 10.694 | ND |
| Esam | Endothelial | 11.623 | 3.184E-04 |
| Flt1 | Endothelial | 9.538 | ND |
| Cldn5 | Endothelial | 15.594 | ND |

fractalkine receptor Cx3cr1 and fat mass response (p = 9.62E–10). Fractalkine (Cx3cl1) is a chemokine, that was recently implicated in diet induced obesity, insulin regulation and promotion of hypothalamic inflammatory response to fatty acids (*Shah et al., 2015*). However, different models of Cx3cr1 knockout mice resulted in variable results on diet induced obesity. The correlation we found is consistent with previous reports that identify a central role for fractalkine receptor Cx3cr1 as a regulator of diet induced obesity and hypothlamic inflammation (*Lee et al., 2013*, *Morari et al., 2014*). Our results also indicate that the expression of Cx3cr1 is affected by genetic background, and suggest that one possible explanation for the heterogeneity in Cx3cr1 knockout results is the different genetic backgrounds used in those studies.

## Discussion

In this work we present a comprehensive picture of the transcriptome of the mouse hypothalamus and its genetic variation and regulation. We identify thousands of new isoforms, and >400 new

genes, and show independent support for these being translated into protein, which further validates our data. Notably, transcription of pseudogenes had been noticed previously, and likely plays a role in gene regulation. Recent shotgun proteomic studies of the human proteome strongly suggest that a sizable fraction of pseudogenes and lncRNAs is translated (*Ji et al., 2015*; *Kim et al., 2014*). The peptide data from our study supports low level translation of processed pseudogenes and is in line with these results.

The hypothalamus is a highly heterogeneous tissue with multiple nuclei and cell types acting in concert. A recent RNA-seq study of fed and food-deprived mice showed that cell type specific transcripts in hypothalamic Agrp and Pomc neurons exhibited specific co-expression networks associated with feeding (*Henry et al., 2015*). In contrast, due to the practical constrains of the study, our data were collected on the entire hypothalamus, and as such would be less sensitive to cell type specific signals. Nevertheless, expression of cell-specific markers and functional molecules showed that our approach recapitulates known correlations between genes and metabolic phenotypes, and identifies new ones. In addition, our data capture the various non neuronal cell types, such as microglia or astrocytes, which are often overlooked in the mostly neuron focused studies of the hypothalamus. These cells are important mediators of hypothalamic inflammation and other processes induced by a high fat diet. Regulation of gene expression in this population impacts every aspect of metabolism. While cell type specific transcriptomics is valuable for understanding cellular processes in hypothalamic neurons, our data provide a robust framework recapitulating transcriptional processes affecting multiple cell populations. Our approach is thus complementary to the cell type-specific transcriptomic efforts.

In this study, we showed extensive genetic regulation of transcription and alternative splicing in the hypothalamus and identified two loci which influence transcription of hundreds of genes in *trans*. In addition, our data indicate that a considerable proportion of the new isoforms and transcripts are significantly correlated to physiological phenotypes. While human studies generally lack the power to address *trans* regulation on a genome-wide scale, the HMDP provides a powerful resource for such analysis. Indeed, we identified two very strong *trans* acting hotspots that seem to harbor major regulators of gene expression in hypothalamus. We suggest that the *trans* effects of genetic variation in these regions are likely mediated by *local* interactions, which is consistent with previously observed cases (*Small et al., 2011*; *Heinig et al., 2010*). Enrichment analyses clearly suggest that each *trans* eQTLs hotspot regulates a set of functionally enriched genes (e.g. the hotspot on chromosome 15 is strongly associated with ion transport in synaptic areas) suggesting a new link between genetic variants in these loci and specific cellular function. We further showed that the genes associated with these hotspots correlate to physiological phenotypes, such as HDL and triglyceride levels providing insight into the mechanism behind correlation of these genotypes with complex traits. The connection between the associated genes and traits does not imply direct causality. For example, ion transport is regulated by circadian rhythm (*Ko et al., 2009*), which in turn is associated with many other aspects of metabolism (*Tsuneki et al., 2016*).

Notably, although several examples of *trans* eQTL hotspots were published and analyzed (*Small et al., 2011*, *Heinig et al., 2010*; *Tian et al., 2016*), authenticity of such hotspots remains somewhat controversial, and *trans* eQTL hotspots may arise due to uncontrolled batch effects (*Breitling et al., 2008*; *Kang et al., 2008*; *Joo et al., 2014*), which are difficult to distinguish from real interactions. To minimize technical batch effects arising from sequencing, we used a round robin approach where each sample was sequenced as part of several pools (see details in Materials and methods). Batch effects may also arise from random environmental exposures, rather than technical sample preparation. In such case, it is unlikely that those effects would be tissue specific or affect only gene level analysis. We found no indication of these *trans* eQTL hotspots in the adipose or liver data from the same cohort of mice (*Parks et al., 2013*), nor were the hotspots detected for isoQTL. Thus, while we cannot exclude the possibility that *trans* eQTL hotspots described in this study may had arisen from unaccounted environmental or technical effects, we believe that this is unlikely, but further molecular studies are required to validate our results.

RNA-Seq allowed us to quantify transcripts at different levels of analysis- from the total expression of all isoforms in a locus, to transcript specific estimates and their combinations. Our analysis showed that regulation of the total number of transcripts from a gene is far more common than isoform specific regulation. Nevertheless, we were able to identify specific interactions at every level of detail we explored, namely eQTLs, isoQTLs, tssQTLs and cdsQTLs. In addition, we identified and

quantified >7000 alternative splicing events affecting >3500 genes in the hypothalamus, and showed that these events are mostly affected by extensive *local* genetic regulation. In many cases of alternative splicing QTLs, the associated variant resulted in an excess of either 0 or 1 splice PSI values, suggesting that variants affecting splicing act as strong determinants rather than weak contributors to a complex trait.

RNA editing is a result of post transcriptional deamination processes whereby an adenosine is converted to inosine (A-to-I), or cytosine to uracil (C-to-U). Both types of RNA editing are mediated by specific enzymes members of the ADAR family facilitate the A-to-I editing which is most commonly observed in neuronal tissues, while Apobec1 mediates the C-to-U editing, and is primarily expressed in the intestine and liver. Editing is a tissue specific process which usually results in modification of only fraction of the transcripts, and therefore can be regarded as a quantitative trait. Previous studies showed that polymorphisms either in the editing enzymes or in the sequence proximal to editing site affect the extent of editing. Specifically, Gu et al recently reported a sequence variant in Apobec1 which affects in *trans* multiple C-to-U editing sites in the mouse liver (*Gu et al., 2015*). As expected from previous studies, A-to-I was by far the most common RNA editing event in our study. Further, our data show that RNA editing occurs at low levels in thousands of sites, but is highly variable. Less than <1% of the sites were consistently detected in the adult mouse brain across >75% HMDP strains. Consistent with the lack of coding genetic variation in the ADAR enzymes in our mouse panel, and with their invariable expression levels, we did not observe any *trans* editing QTL that would affect editing levels of multiple sites. However, we found a few local associations that seem to affect editing of specific sites. For example a strong *local* editing QTL (p<e−7) was observed for one of the editing sites in the *Ociad1* gene (also known as *Asrij*). This editing QTL is likely due to a sequence variant in or near the *Ociad1* gene. Notably, *Ociad1* is expressed in stem cells, where it regulates pleuropotency via the JAK/STAT pathway (*Sinha et al., 2013*). Editing of *Ociad1* or its expression in neuronal tissue was not reported before.

Another significant aspect of our data relates to long non-coding RNAs. LincRNAs have been shown to play an important role in various cellar functions. Recent genome annotations include thousands of known lincRNAs, yet most of them remain functionally uncharacterized, and only a few studies have examined the genetic control of their expression in detail. Moreover, lincRNAs are often expressed in a tissue specific manner, and therefore are not readily identifiable from general expression datasets. In this study, we detected expression of 381 known and novel lincRNA isoforms, and also identified 129 novel, potentially hypothalamus- specific lincRNAs. Our data indicate that lincRNAs are subject to similar variation in expression, and exhibit similar overall genetic control as the coding genes. Specific lincRNAs have been implicated in a variety of phenotypes (*Guttman et al., 2011*; *Huarte et al., 2010*; *Kumar et al., 2013*), and our data indicate strong correlations between some of the hypothalamic lincRNAs and metabolic phenotypes, such as body weight.

The hypothalamus is a very heterogeneous tissue, and one of the major drawbacks of our analysis is that we used whole hypothalamus, rather than dissecting specific nuclei. This limitation stems from practical considerations – meaningful expression QTL analysis requires sacrifice of hundreds of mice, while the dissection of specific hypothalamic nuclei is delicate and time consuming and thus was not feasible within the constraints of this study. Still, this shortcoming is likely to limit our power to detect meaningful associations, rather than introducing spurious ones. Moreover, since our study mostly focuses on genetic regulation of transcription, which was shown to be largely shared among tissues and cell types, we believe that the data presented in this work are not substantially confounded by heterogeneity.

To summarize, our data fill a substantial gap and will be useful to the research community. The hypothalamus is composed of multiple nuclei, which are distinct in morphology and function, and many laboratories focus on disentangling the complex interactions that ultimately affect metabolism and behavior. However, genome wide transcriptome analysis of this tissue has not been published, and genetic regulation of transcription as well as tissue specific transcripts remains largely obscure. We believe that our study is complimentary to physiological studies and will facilitate research into the crosstalk between the brain and other metabolic tissues. All of the expression data described in this paper are publicly available from NCBI archives GEO (GSE79551) and SRA (project number PRJNA314533).

## Materials and methods

### Samples, library preparation and sequencing

Altogether we sequenced 285 samples, from 99 strains of the HMDP. A total of 87 strains had 3 samples per strain, 11 strains had 2 samples per strain, and 2 strains had 1 sample per strain (a detailed list is in *Supplementary file 1*). RNA was extracted using Qiazol followed by miRNAeasy kit from Qiagen (RRID:SCR_008539). Unstranded mRNA libraries were prepared by the UCLA Neuroscience Genomics Core with Illumina standard kits (TruSeq v3) according to standard protocols. All samples were barcoded, and sequenced with ~18 samples per lane, with HiSeq2000 using 50 bp paired-end sequencing protocol. A round robin design was implemented such that biological replicates were sequenced on different lanes, and each sample was part of more than one sequencing pool. Samples were demultiplexed by sequencing facility, forward and reverse read fastq files were supplied for each sample.

### Data quality control (QC) and mapping

Read QC was done using FastQC (RRID:SCR_005539) in batch mode. The samples had excellent quality, with all bases exceeding median quality score of 28, and >98% of the sequences with a mean quality score >28. The average number of reads per sample was 26.3 M. All reads were passed to mapping as is, without trimming or filtering. Samples were mapped to the mm10 genome using STAR aligner version 2.3.1 (https://github.com/alexdobin/STAR/releases/tag/STAR_2.3.1z9). The mm10 sequence was downloaded from http://cufflinks.cbcb.umd.edu/igenomes.html (UCSC annotation files). Reference sequences were built using known splice junctions (–*sjdbGTFfile* option) from known genes annotation file. Mapping was performed allowing up to 3 mismatches per read (–*outFilterMismatchNmax 3*), removing non canonical un-annotated junctions (–*outFilterIntronMotifs RemoveNoncanonicalUnannotated*) and allowing up to 10 multiple mappings per read (–*outFilterMultimapNmax 10*). Alignment files from all samples of the same strain were merged into one alignment file per strain using the 'merge' function of the samtools package. On average 94.1% of the reads mapped, and of the mapped reads 96.4% mapped uniquely. STAR also detected on average 2.8 M splices per sample, with 98% of them previously annotated. Detailed mapping counts for each sample are in *Supplementary file 1*.

### Transcript assembly and quantification

Our pipeline is summarized in *Figure 8*. For the purpose of transcript assembly, sample specific alignment files were pooled into unified BAM alignment files by strain which were then sorted and indexed using samtools. Transcript assembly was done on each strain specific alignment file with Cufflinks v2.2.0., with mouse genome version mm10, and gtf file of known transcripts as a reference guide (-*g* option, reference file downloaded from UCSC genome browser), together with bias and

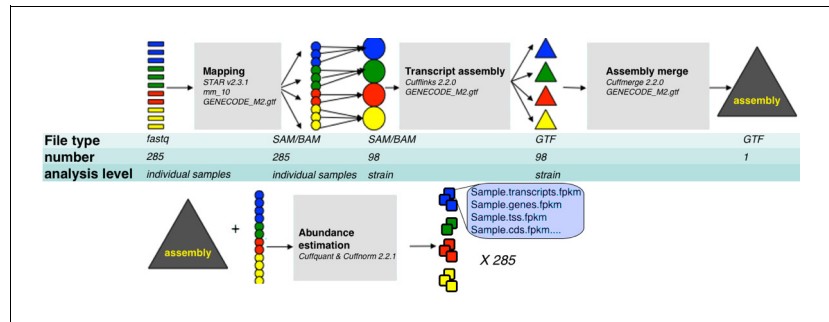

**Figure 8.** RNA-Seq analysis framework. General workflow used for analysis of RNA-Seq data in this study. Initial demultiplexed samples (fastq files) were aligned to the mouse genome with STAR, merged in one file per strain, and transcripts assembeled with cufflings. The resulting assembly files (one from each strain) were merged with GENECODE M2 annotation into unified assembly. The abundance of each transcript in the unified assembly was estimated in sample specific alignment files.

multimap option corrections (-*b* and –*u* respectively). This step resulted in 99 strain specific transcript assemblies. We then used Cuffmerge to compare those assemblies to GENCODE M2 annotation (http://www.gencodegenes.org/), and to consolidate them into one unified assembly file representing all transcripts from GENCODE M2 and from the strain specific assemblies (merged.gtf). To obtain FPKM expression values, we run Cuffquant and Cuffnorm v2.2.1 with default parameters, using the sample specific alignment files and unified assembly (merged.gtf) file.

## Filtering of the expressed features for subsequent analyses

Cufflinks assembly and abundance estimation results in assessment of gene expression at multiple levels of genomic resolution - transcripts, transcription start sites, coding sequence and loci (genes). We implemented 3 filtering steps:

First, we removed transcripts that are likely to be an assembly or sequencing artifacts, based on following criteria:

Comparison to GENECODE M2 indicated that this is not a known transcript (class code '='), new isoform of a known gene (class code 'j') or novel intergenic transcript (class code 'u').

b. Novel isoforms and intergenic transcripts which expression was <1e-6 FPKM in all samples.

Loci and transcription start sites associated exclusively with transcripts removed in steps 1 and 2.

This filtering step removed 7% of the transcripts, with the other 93% (535,316 features) treated as potentially true.

II. While we consider that all these features may potentially represent true splicing and transcription events, only features that show sufficient variation and expressed at appreciable level are useful in eQTL analysis. Further, many of the potentially new features were expressed either at very low levels or in a small number of samples. Therefore, at the second filtering step we implemented detection and expression thresholds (FPKM>0 in >=100 samples, and FPKM>1 in >=50 samples) to focus on ~100,000 expressed features that are more likely to produce meaningful mapping results in the HMDP panel.

Third, expression values (FPKM) of transcription start sites, coding sequences and genes are a sum of expression values of transcripts associated with these features. Consequently, if a locus, transcription start site or coding sequence is associated with only one transcript, the expression information of that feature is identical and thus redundant to the respective transcript. Therefore, such features were removed from eQTL mapping.

Altogether, this filtering reduced the number of explored features to ~ 93,000.

## Allele-specific expression and cis eQTL

All features that passed the described above filtering criteria, were analyzed for eQTLs using expression estimates in 282 individual mice representing 100 strains, 193,400 SNPs and the fastLMM algorithm using an appropriate kinship matrix that accounts for the HMDP population structure. *Cis* acting eQTLs were identified by allele expression (ASE) as described previously (*Hasin-Brumshtein et al., 2014*). Briefly, allele-specific counts at each exonic SNP, filtered for quality control criteria, were summed in a genespecific manner. The resulting counts were normalized and analyzed for differential expression between the alleles using the DEseq package.

## Identification of RNA editing sites

To identify RNA editing sites, we re-aligned the reads to the mouse genome (mm10) and transcriptome using the single nucleotide variant-sensitive aligner RASER (*Ahn and Xiao, 2015*) with the parameters –m 0.06 and –b 0.03. We then detected likely mismatch positions considering the read sequence quality score at that position and its position within in a read. We further required potential editing sites to be covered by >=5 total and >=3 edited reads and excluded positions that overlap SNPs or are located in homopolymers, simple repeats, and within 4nts of splice junctions (*Lee et al., 2013*).

## LC-MS proteomics

Proteomic data were collected on 110 samples representing the 99 strains used in this study, plus 10 C57Bl/6 samples. All samples were independent biological replicates of the strains used in this study fed on the high fat, high sugar diet, and not the same hypothalami used for RNA extraction.

## Sample preparation

The preparation of 110 mouse hypothalamus samples for LC-MS/MS analysis was performed according to the protocol described before (*Ghazalpour et al., 2011*). Briefly, the samples were split evenly across two 96-well plates, and homogenized using TissueLyser (QIAGEN, Hilden, Germany) in (8 M urea, 50 mM $NH_4HCO_3$ pH 7.8 and 10 mM DTT) at 20 Hz for 2 min with two repetitions. Upon denaturation for 1 hr at 37°C the extracted protein concentrations were measured using coomassie assay and readjusted using aforementioned denaturation buffer to 1.5 mg/ml. Equal aliquots of 150 µg were taken further into cysteine alkylation and trypsin digestion steps. After purification of peptides by solid phase extraction using C18 columns the average recovered peptide amount was 68 µg. The resulting peptide solutions were normalized to 0.5 µg/µL for LC-MS/MS analysis

## Instrumental analysis

Samples were analyzed on an Orbitrap Velos mass spectrometer (Thermo Fisher, Waltham, MA) that was interfaced with a 75 µm i.d.×65 cm long LC column packed with 3 µm Jupiter C18 particles (Phenomenex) and a 5 µL injection loop. The mobile phase solvents consisted of (A) 0.2% acetic acid and 0.05% TFA in water and (B) 0.1% TFA in 90% acetonitrile. An exponential gradient was used for the separation, which started with 100% A and gradually increased to 60% B over 100 min.

## Data analysis

The acquired MS/MS spectra datasets were preprocessed with DeconMSn [18304935] and DtaRefinery [20019053] software followed by spectra interpretation using MSGFplus [25358478] software by matching against custom protein fasta file build based on RNA-Seq data. The fasta file included sequences of known genes, newly discovered isoforms and novel genes. The MSGFplus MS/MS search settings were as follows: tryptic peptides only, 10 ppm parent ion mass tolerance, static cysteine carbamylation and maximum charge state 4+. The resulting mzIdentML files were analyzed using MSnID Bioconductor package [http://www.bioconductor.org/packages/release/bioc/html/MSnID.html] to confidently isolate identifications of know genes, isoforms and novel genes. The prior probabilities that the know genes, isoforms and novel genes are real and exist in the protein form are substantially different. Therefore the corresponding identifications were considered separately employing a process effectively emulating the proposed earlier cascade search [26084232]. Briefly, in the first pass we considered peptide identifications matching only to the protein sequences of the known genes (highest prior probability). The filtering criteria on peptide-to-spectrum matching (Spectrum E-Value <9.4e-11 and parent ion mass measurement tolerance <2.4 ppm) were optimized to achieve maximum peptide identifications whilst not exceeding 1% FDR based on reverse sequence identifications. Next, the spectra that match confidently identified 9910 peptide sequences of the known genes were removed from further consideration. In the next round we applied filtering criteria (Spectrum E-Value <2.5e–13 and parent ion mass measurement tolerance <10 ppm) that allowed to confidently identify 236 unique peptide sequences matching to isoforms. After further removing peptides matching to isoforms and applying 1% FDR optimized criteria (Spectrum E-Value <2.3e–9 and parent ion mass measurement tolerance <0.2 ppm) we identified 20 peptide sequences matching novel genes.

## Additional information

### Funding

| Funder | Grant reference number | Author |
| --- | --- | --- |
| National Institutes of Health | R01GM098273 | Yehudit Hasin-Brumshtein<br>Arshad H Khan<br>Calvin Pan<br>Vladislav A Petyuk<br>Paul D Piehowski<br>Richard D Smith<br>Aldons J Lusis<br>Desmond J Smith |

| National Institute of General Medical Sciences | GM103493 | Vladislav A Petyuk Paul D Piehowski Richard D Smith |
|---|---|---|
| W.R. Wiley Environmental Molecular Science Laboratory | DE-AC05-76RLO-1830 | Vladislav A Petyuk Paul D Piehowski Richard D Smith |
| National Institutes of Health | R01HG006264 | Anneke Brümmer Xinshu Xiao |

The funders had no role in study design, data collection and interpretation, or the decision to submit the work for publication.

## Author contributions

YH-B, Conception and design, Acquisition of data, Analysis and interpretation of data, Drafting or revising the article; AHK, Conception and design, Acquisition of data, Analysis and interpretation of data; FH, CP, AB, XX, Analysis and interpretation of data; BWP, PDP, Acquisition of data; VAP, Acquisition of data, Analysis and interpretation of data; MP, Conception and design; EE, Conception and design, Analysis and interpretation of data; RDS, Conception and design, Acquisition of data; AJL, Conception and design, Analysis and interpretation of data, Drafting or revising the article, Contributed unpublished essential data or reagents; DJS, Conception and design, Acquisition of data, Analysis and interpretation of data, Drafting or revising the article, Contributed unpublished essential data or reagents

## Author ORCIDs

Yehudit Hasin-Brumshtein, http://orcid.org/0000-0001-7528-603X

## Ethics

Animal experimentation: The animal protocol for the study was approved by the Institutional Animal Care and Use Committee (IACUC) at University of California, Los Angeles.

# Additional files

## Supplementary files

• Supplementary file 1. Annotations of novel genes and samples. (A) Annotation of novel genes and transcripts Basic annotation data of all transcripts classified as 'new locus' (class code = u).Data includes tracking ID of the transcript and gene, locus position, number of samples where transcript is detected (FPKM>1e-6) and number of samples where transcript is expressed (FPKM >1), mean expression across all samples and if gene arises from uniquely mapped reads. (B) Annotation of detected peptides Basic annotation data of all peptides detected by LC-MS proteomics. Data includes peptide sequence, tracking ids of matching transcripts (known, novel isoforms and novel genes, based on Cufflinks class code classification), number and list of strains in which the peptide was detected, and number of transcripts the peptide may be attributed to. (C) Sample description. Technical metadata pertaining to samples used in this study, including NCBI SRA accession numbers, RNA-Seq QC data (number of reads, mapped reads, detected junctions), and mouse id and strain.

• Supplementary file 2. Trans eQTL hotspots. Includes counts of associated genes in 100 kb windows, and for each of the hotspots: a list of associated genes, interaction p values, list of local eQTLs for the top SNP and DAVID enrichment annotation of the genes associated in trans.

• Supplementary file 3. Trans eQTL hotspots - gene counts, functional enrichment and local QTLs. (A) Gene-trait correlations Top known 500 genes associated with each phenotype in HMDP.Data is aggregated in table form crossing all genes with all traits, thus not all gene-trait pairs are significant. 'Inf 'indicates not significant interactions. Numeric values indicate p value of association, with 1E-3 correlating to 5% FDR threshold based on permutations. (B) Novel genes are associated with metabolic traits Data table with all detected associations between novel genes and the 150 phenotypes assessed in HMDP.

**Major datasets**

The following datasets were generated:

| Author(s) | Year | Dataset title | Dataset URL | Database, license, and accessibility information |
|---|---|---|---|---|
| Yehudit Hasin-Brumshtein, Arshad H Khan, Farhad Hormozdiari, Calvin Pan, Brian W Parks, Vladislav A Petyuk, Paul D Piehowski, Anneke Bruemmer, Matteo Pellegrini, Xinshu Xiao, Eleazar Eskin, Richard D Smith, Aldons J Lusis, Desmond J Smith | 2016 | Data from: Hypothalamic transcriptomes of 99 mouse strains reveal trans eQTL hotspots, splicing QTLs and novel non-coding genes | http://doi.org/10.5061/dryad.vm525 | Available at Dryad Digital Repository under a CC0 Public Domain Dedication |
| Lusis AJ, Smith D, Hasin-Brumshtein Y | 2016 | Hypothalamic transcriptome of male mice on high fat diet, from 99 strains | http://www.ncbi.nlm.nih.gov/geo/query/acc.cgi?acc=GSE79551 | Publicly available at the NCBI Gene Expression Omnibus (accession no: GSE79551) |

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
