## [Decision Letter]

[Editors’ note: this article was originally rejected after discussions between the reviewers, but the authors were invited to resubmit after an appeal against the decision.]

Thank you for submitting your work entitled "Hypothalamic transcriptomes of 99 mouse strains reveal trans eQTL hotspots, splicing QTLs and novel non-coding genes" for consideration by *eLife*. Your article has been reviewed by three peer reviewers, and the evaluation has been overseen by a Reviewing Editor and Mark McCarthy as the Senior Editor. One of the three reviewers has agreed to reveal her identity: Penelope Bonnen (Reviewer #1).

Our decision has been reached after consultation between the reviewers. Based on these discussions and the individual reviews below, we regret to inform you that your work will not be considered further for publication in *eLife*.

While all of the reviewers found your data to be of potential interest, they all raised substantive concerns regarding both study design and data presentation. The consensus was that addressing all of these issues would entail substantial work. As you may know, one of the goals of *eLife* is to avoid protracted cycles of "long-loop" revision, and this means that we try to avoid any recommendations for revision that are likely to take longer than a few weeks.

*Reviewer #1:*

1) Trans eQTL hotspots analysis is nicely done and the authors relate two well-supported and interesting examples of trans eQTLs. Two comments for this section. Firstly, the method used to identify regions of trans eQTLs was blind to LD; a simple distance (100 kb) based bin was used and the number of trans acting SNPs counted. The authors relate that a 'hotspot' of trans acting SNPs contained several SNPs in LD with each other. This approach is valid, but may skew results to identify regions with a larger number of trans acting SNPs because of the underlying LD structure. In this case LD helps to identify these regions. However, there may be regions with less LD but still strong trans acting SNPs; using the method these regions will be de-emphasized, not due to the strength of trans acting elements but rather due to LD architecture of the genome. The authors may wish to mention this. Second, poorly annotated regions of the genome are not devoid of functional elements and it may be a bit of an overstatement to say the eQTL SNPs in these regions are unlikely to be causative.

2) Which lincRNAs were included in the association testing between lincRNAs and HMDP phenotypes? Were both the 381 plus 129 novel tested? In particular it would be interesting to clarify if any of the 129 newly identified lincRNAs associate with HMDP phenotypes. This reviewer found this sentence a little confusing 'none of these lincRNA were previously reported in the hypothalmus'; does this mean they all belonged to the group of 129 novel lincRNAs? Were these 129 totally novel or simply not previously known to be expressed in the hypothalamus? Some clarification would be helpful and would emphasize the findings that are particular to this study.

*Reviewer #1 (Additional data files and statistical comments):*

Please include the top 500 transcripts that associate with a phenotype in supplemental materials listed along with transcript ID, phenotype, and p-value. These are referred to in the first paragraph of the subsection “Association of hypothalamic expression and phenotypes” and are a valuable work product of this study. As such readers may benefit from this information.

*Reviewer #2:*

This manuscript focuses on a detailed analysis of hypothalamic transcriptomes from 99 mouse strains to among many things identify eQTL, spicing eQTL and novel non-coding genes and their relationship to metabolic and cardiovascular traits. Many of the aspects of the relevance or tissue-specific findings pertaining to hypothalamus were not well clarified leading to a concern with the overall study design. Many of the analyses suffer from potential interpretation or technical flaws limiting my enthusiasm.

A key feature of the manuscript is the identification of numerous novel isoforms and genes. It is an exceptional feature that the authors are able to confirm their data using peptide data. However, most of these are pseudogenes when manual characterized against NCBI and UCSC. Is this simply a deficiency in the GENCODE M2 annotation? Further, can the authors discount mismapped reads from highly homologous sequences to these pseudogenes. Key aspects of this pipeline are not well presented and I am very skeptical of the "novelty" of the set of genes they have found or even their relevance to the hypothalamus in specific let alone any hypothalamus specific function. Criteria for tissue-specificity should further be better described.

The distal eQTL information provides no extra information. Indeed, the authors themselves describe it as most likely a local signal.

The correlation of ASE and eQTL is a technical feature that has been reported in several other papers. It shouldn't be listed as a major feature of the manuscript and would be better as supplemental information.

The authors derive their own ASE approach using DEseq. This is not a conventional approach for this tool. Other published ASE-specific software exists. Can the authors explain why they did it this way?

How correlated are ASE sites within a gene? Does this support site aggregation for a gene-centric approach?

Trans-eQTLs can be spurious due to the chance correlation of a genetic marker with technical factors such as batch or biological factors such as sex or BMI. The authors do not provide enough detail/experimentation as to make these effects believable as true trans-eQTLs compared to spurious correlations.

Furthermore, for RNA-seq data many 1-1 trans-eQTLs can be the product of mismapping and are clearly identified when looking at the distribution of reads across a gene model in a tool like IGV.

The comment on rs31703733 is very speculative. The authors should prove that this is the case otherwise, I am not sure there is anything that can be definitely said about trans-eQTLs in this manuscript.

Some of the statements in the manuscript could benefit from more formally described multiple testing correction or an FDR. For instance "35% of lincRNA.… significantly (p<1e-3) correlate to at least one phenotype in the HMDP". If I do some rudimentary correction assuming 150 traits, I am not sure this is a significant finding. This is an issue for the trait analysis of the novel transcripts too.

How believable are the non A-to-G modification? Is 30% an expected number?

*Reviewer #3:*

In this manuscript, Hasin-Brumshtein and coauthors provide a systems-level analysis of hypothalamic gene expression, its regulation by genetic elements, and its association with metabolic phenotypes. The hypothalamus is composed of diverse and distributed neuronal subpopulations that control multiple aspects of metabolic homeostasis. Here, the authors test the hypothesis that differences in hypothalamic gene expression in a hybrid mouse diversity panel (HMDP) may influence the sensitivity of particular strains to phenotypes associated with diet-induced obesity. The authors provide two specific examples from their data of transcripts (1 annotated and 1 lincRNA) whose expression correlates with a specific metabolic endpoint (Figure 4 and Figure 5). To increase confidence in the analysis, the authors should validate the hypothalamic expression differences between strains for a handful of these transcripts by in situ hybridization.

Overall, the lack of functional validation of candidate genes with phenotypic outcomes severely limits this as a resource, and greatly diminishes the impact of this study. Nonetheless, this study does begin offering a detailed mapping of genomic regions underlying differential gene expression and RNA processing events in the hypothalamus, which with additional work could enhance our functional understanding of the hypothalamus. However, in its present form, it is not accessible to the research community working in the hypothalamus.

Hasin-Brumshtein et al. characterize the expression of both annotated and novel genes, transcript variants, and lincRNAs from the hypothalamus of mice representing 99 strains from the HMDP. The effort invested in obtaining and analyzing both genomic and proteomic data at this scale is impressive. However, profiling data presented would benefit from a more thorough description of mouse husbandry conditions prior to tissue collection. Were the mice raised on the same chow diet used by Parks et al. (2013) to establish the phenotypic responses of mice in the HMDP to high fat, high sucrose diet? In fact, the authors' description in the Introduction suggests that RNA-Seq was performed on "mice… fed a high fat high sugar diet." If so, it is unclear if the observed expression differences across the mouse strains drive their unique phenotypic responses to dietary challenge or if these differences are a response to such a challenge. Similarly, since RNA-Seq was performed on as few as 1 subject per strain, the authors should elaborate on steps taken to mitigate other environmental factors that might have influenced gene expression in the mice. Why not increase the power of this study by increasing the numbers on the most commonly used strains with the greatest divergence in metabolic outcomes?

The authors do acknowledge one major weakness which is the pan-hypothalamic approach, thus reducing their ability to identify genotype/phenotype associations with transcripts (Discussion, sixth paragraph). However, the statement at the end of that paragraph: "transcription… was shown to be largely shared among tissues and cell types" is confusing. If this is true, would not the small differences in expression between subpopulations of neurons be the most important in determining their functional specificity and in turn contribute to phenotypic differences? Such small differences are the least likely to be identified by analyzing the whole hypothalamus. Alternatively, the differences could arise from altered synaptic connectivity, which would also be difficult to resolve by expression profiling. This is an important point that is never discussed.

In addition, given that the hypothalamus is a critical region for many important sex-dependent metabolic and behavioral outcomes due to signaling from sexually dimorphic subsets of neurons, one wants to know how these data compare between male and females.

[Editors’ note: what now follows is the decision letter after the authors submitted for further consideration.]

Thank you for resubmitting your work entitled "Hypothalamic transcriptomes of 99 mouse strains reveal trans eQTL hotspots, splicing QTLs and novel non-coding genes" for further consideration at *eLife*. We were receptive to the basis of your appeal, and the revised article has been favorably evaluated by Mark McCarthy (Senior editor), the original Reviewing editor, and by the more critical of the previous sets of reviewers.

All of us are positive about your revision. However, there are a couple of issues that remain to be addressed before acceptance, as outlined below:

1) The caveat the authors have added to the text is acceptable but the authors should look at Gencode M4 and see if there are overlaps with their genes. This annotation has been available for 2.5 years and is the most current mouse annotation file. M2 dates back to 2013. The rationale that annotation changes too rapidly ("month-to-month") is not supported by the Gencode release dates and since the novelty of these transcripts is a major feature, it should be put in line with the latest (2.5y old) annotation. One could simply overlap novel locations with bed files of latest annotations at a minimum.

2) Another issue that remains is the trans-eQTL results as any SNP correlated with a batch effect can appear as a trans-eQTL hotspot. The trans-eqtl on chr15 relevant to ion transport might be reflecting circadian rhythm. http://www.ncbi.nlm.nih.gov/pmc/articles/PMC2819050/

The authors should explore this issue in more detail or properly caveat it too.

---

## [Author Response]

[Editors’ note: the author responses to the first round of peer review follow.]

*While all of the reviewers found your data to be of potential interest, they all raised substantive concerns regarding both study design and data presentation. The consensus was that addressing all of these issues would entail substantial work. As you may know, one of the goals of eLife is to avoid protracted cycles of "long-loop" revision, and this means that we try to avoid any recommendations for revision that are likely to take longer than a few weeks.*

*Reviewer #1:*

*1) Trans eQTL hotspots analysis is nicely done and the authors relate two well-supported and interesting examples of trans eQTLs. Two comments for this section. Firstly, the method used to identify regions of trans eQTLs was blind to LD; a simple distance (100 kb) based bin was used and the number of trans acting SNPs counted. The authors relate that a 'hotspot' of trans acting SNPs contained several SNPs in LD with each other. This approach is valid, but may skew results to identify regions with a larger number of trans acting SNPs because of the underlying LD structure. In this case LD helps to identify these regions. However, there may be regions with less LD but still strong trans acting SNPs; using the method these regions will be de-emphasized, not due to the strength of trans acting elements but rather due to LD architecture of the genome. The authors may wish to mention this. Second, poorly annotated regions of the genome are not devoid of functional elements and it may be a bit of an overstatement to say the eQTL SNPs in these regions are unlikely to be causative.*

We are glad that reviewer found our trans eQTL analysis interesting. Regarding the LD comments, we agree that stronger and longer LD will increase the number of SNPs associated in trans, however it is not likely to increase the number of genes associated with each bin, since essentially linked SNPs will be capturing the same interactions. In our analysis (Figure 4) we counted the number of different genes associated with any SNP in the bin, and not (as reviewer seems to suggest) the number of trans acting SNPs, thus we believe LD does not significantly impact our identification of hotspots. We are sorry for this confusion, and rewrote the relevant sentences to better clarify our approach (subsection “Genetic regulation of gene expression”), and made it clearer in the Discussion as well.

*2) Which lincRNAs were included in the association testing between lincRNAs and HMDP phenotypes? Were both the 381 plus 129 novel tested? In particular it would be interesting to clarify if any of the 129 newly identified lincRNAs associate with HMDP phenotypes. This reviewer found this sentence a little confusing 'none of these lincRNA were previously reported in the hypothalmus'; does this mean they all belonged to the group of 129 novel lincRNAs? Were these 129 totally novel or simply not previously known to be expressed in the hypothalamus? Some clarification would be helpful and would emphasize the findings that are particular to this study.*

We thank the reviewer for the comments, and have rewritten the section to clarify these points (subsection “Long non-coding RNAs in the mouse hypothalamus”).

*Reviewer #1 (Additional data files and statistical comments):*

*Please include the top 500 transcripts that associate with a phenotype in supplemental materials listed along with transcript ID, phenotype, and p-value. These are referred to in the first paragraph of the subsection “Association of hypothalamic expression and phenotypes” and are a valuable work product of this study. As such readers may benefit from this information.*

We have added the requested file to the supplementary material (Supplementary file 5).

*Reviewer #2:*

*This manuscript focuses on a detailed analysis of hypothalamic transcriptomes from 99 mouse strains to among many things identify eQTL, spicing eQTL and novel non-coding genes and their relationship to metabolic and cardiovascular traits. Many of the aspects of the relevance or tissue-specific findings pertaining to hypothalamus were not well clarified leading to a concern with the overall study design. Many of the analyses suffer from potential interpretation or technical flaws limiting my enthusiasm.*

*A key feature of the manuscript is the identification of numerous novel isoforms and genes. It is an exceptional feature that the authors are able to confirm their data using peptide data. However, most of these are pseudogenes when manual characterized against NCBI and UCSC. Is this simply a deficiency in the GENCODE M2 annotation? Further, can the authors discount mismapped reads from highly homologous sequences to these pseudogenes. Key aspects of this pipeline are not well presented and I am very skeptical of the "novelty" of the set of genes they have found or even their relevance to the hypothalamus in specific let alone any hypothalamus specific function. Criteria for tissue-specificity should further be better described.*

We thank the reviewer for this comment. In the manuscript we present several types of analysis suggesting that the genomic properties of the novel genes, as identified compared to GENCODE M2, are similar to the genomic properties of known non coding transcripts. Specifically, distributions of their potential reading frame length and their splicing pattern indicate that for the most part the novel transcripts would be expected to be non coding and therefore not expected in the peptide data. Thus we did not find it surprising that the very few peptides matching exclusively novel genes can be attributed to pseudogenes. In our opinion this does not imply that the majority of novel transcribed genes are pseudogenes rather that they are probably not translated. We have made this point more clearly in the revised manuscript (subsection “Hypothalamic gene expression and proteomic data reveals multiple new isoforms and novel genes”, third paragraph).

Regarding the reviewers question whether this presents a deficiency in M2 annotation, we would like to add that unlike genome sequence versions, genome annotations tend to develop quite rapidly. For some time GENCODE has released a new set of annotations for the mouse genome every few months. It is impractical to reanalyze the entire dataset every time a new annotation is released. We therefore focused our analysis on M2, which was the annotation we used for our genome indexing and read mapping.

The reviewer raised a question as to whether the expression of novel genes can be attributed to multi-mapped reads. We addressed that question by re-estimating abundances in 3 C57BL/6 samples, using uniquely mapped reads only. Indeed, some of the novel genes (and known genes) show low uniqueness. However, the peptide data supports expression of both uniquely mapped genes and those that arise from multimapping. This observation indicates that a high percentage of multimapping does not necessarily equate to an artifact. We therefore added uniqueness of mapping as an annotation of novel genes and expanded the relevant text to include this analysis (subsection “Hypothalamic gene expression and proteomic data reveals multiple new isoforms and novel genes”, last paragraph and [Supplementary-material SD1-data]).

We agree that tissue specificity of these transcripts cannot be defined based on our data alone, and intend to suggest it as one of the possible explanations. We thank the reviewer for raising this concern, and have rephrased that part of the manuscript to better reflect other potential explanations (Results).

We also thank the reviewer for pointing out that the pipeline is not described in sufficient detail. We have now added a more detailed description, supplementing the already described software versions and specific command lines in the Materials and methods.

*The distal eQTL information provides no extra information. Indeed, the authors themselves describe it as most likely a local signal.*

We agree with the reviewer that distal eQTLs are redundant to local. Distal eQTLs are only mentioned in our manuscript for the sake of completeness (how we selected the eQTLs for analysis). The distal loci were completely excluded from any QTL analysis. We now made this point clearer (Results).

*The correlation of ASE and eQTL is a technical feature that has been reported in several other papers. It shouldn't be listed as a major feature of the manuscript and would be better as supplemental information.*

*The authors derive their own ASE approach using DEseq. This is not a conventional approach for this tool. Other published ASE-specific software exists. Can the authors explain why they did it this way?*

*How correlated are ASE sites within a gene? Does this support site aggregation for a gene-centric approach?*

We agree with the reviewer that ASE is secondary to our results and have moved the ASE analysis to supplementary material. Concerning our choice of method for ASE analysis, we would like to point out that while we indeed created this approach to ASE, the method was not developed for this study and is not novel. In fact, we published this approach in BMC Genomics (Hasin-Brumshtein et al. 2014), with full details and extensive analysis of concordance of ASE within biologically relevant units (exons, transcripts, and genes). The paper also showed that aggregation of ASE signal over haplotype improves specificity of the results (as compared to single SNP). Therefore, since this approach was already developed in our laboratory, published in peer reviewed journal and readily accessible, we chose to use it for the ASE analysis in this paper as well.

*Trans-eQTLs can be spurious due to the chance correlation of a genetic marker with technical factors such as batch or biological factors such as sex or BMI. The authors do not provide enough detail/experimentation as to make these effects believable as true trans-eQTLs compared to spurious correlations.*

*Furthermore, for RNA-seq data many 1-1 trans-eQTLs can be the product of mismapping and are clearly identified when looking at the distribution of reads across a gene model in a tool like IGV.*

*The comment on rs31703733 is very speculative. The authors should prove that this is the case otherwise, I am not sure there is anything that can be definitely said about trans-eQTLs in this manuscript.*

We concur that many of the 1-1 trans interactions may represent spurious results, even when stringent thresholds are used. However, trans eQTL hotspots are much less likely to represent a technical artifact, or a spurious association since that possibility would require the coincidence of multiple artifacts in non-random manner. Thus we only discuss in depth the two trans eQTL hotspots, and do not consider any 1-1 trans eQTLs.

Mapping artifacts are not generally random and might result in false positive signals. Indeed, as the reviewer suggested, it is possible to inspect any given gene for mapping artifacts in IGV or other visualization tools. However, we find it impractical to go manually through the 900 genes associated with one of the 2 hotspots in this way. Thus to address this issue, we assessed whether mapping artifacts or multi mappers significantly contribute to expression levels of the genes in trans eQTL hotspots. To this end we re-estimated the expression of all transcripts in 3 samples, using only high quality mapping scores (mapping quality =255, corresponding to the highest mapping score and unique mapping). Expression levels estimated by this procedure did not change much (correlation values were >0.96 for all genes) showing that mapping artifacts are not likely to account for the variation in expression levels of the trans eQTL hotspots.

The reviewer raises an important concern about potential confounding factors such as sex, BMI and batch effects. First we would like to point out that all of the mice used in this study were males, of the same age and treated with exactly the same protocol. Thus we consider sex age and treatment not to be major confounders of trans-eQTL mapping. Batch effects can arise from two sources in our data, one is the sequencing process, and the other is mice housing conditions. To minimize the batch effects of sequencing we used a round robin design, where we distributed every sample over 3-4 lanes of sequencing, and biological replicates were largely distributed over different lanes, such that no individual strain data was the result of a single lane sequencing. While ideally we would like to pool all samples into one library and sequence that library as many times as need to achieve desired coverage, this was not technically possible because the number of available barcodes is smaller than the number of samples. The other source of batch effects may arise from mouse housing conditions, although great care was taken to match these as well as possible. Still, mice are housed in cages by strain, and work load in a large study like this is invariably distributed over a period of time, thus not all cages are treated on exactly the same days. These factors typically are considered minor in mouse eQTL studies compared to the effects of genetic variance on gene expression, and cannot be fully addressed in a practical manner for eQTL mapping.

Finally the reviewer points out that location of rs31703733 in an enhancer specific to neuronal cells is not strong evidence for its potentially causative role in changes in gene expression. We agree that this is a weak evidence, however we believe that combining several lines of evidence is an important aspect of using our data as resource. We therefore did not remove that point, but rewrote the Results and Discussion in a more reserved manner to convey the issues more clearly (subsection “Trans eQTL hotspots”, last paragraph).

*Some of the statements in the manuscript could benefit from more formally described multiple testing correction or an FDR. For instance "35% of lincRNA.… significantly (p<1e-3) correlate to at least one phenotype in the HMDP". If I do some rudimentary correction assuming 150 traits, I am not sure this is a significant finding. This is an issue for the trait analysis of the novel transcripts too.*

We thank the reviewer for pointing out the need for clarification. Association analysis was performed using linear mixed models (Fast-LMM, Lippert et al., 2011), which corrects for the population structure in our data. Statistical thresholds used in this work were determined as 5% FDR by permutations and modeling in previous eQTL and association studies conducted within HMDP (Parks et al. 2015, Bennet et al. 2010). We now added this point explicitly to the Results section to better explain our choice. Importantly, since many of the metabolic phenotypes are not independent, and are significantly correlated, the rudimentary approach suggested by the reviewer would be overly stringent.

*How believable are the non A-to-G modification? Is 30% an expected number?*

A-to-G modifications and C-to-T modifications (another canonical type of editing catalyzed by APOBEC), together make up ~78% of the detected RNA-DNA differences (RDD) when requiring a minimum of 5 total and 3 edited reads covering the editing position.

Since the paired-end RNA-seq reads are non-strand specific and we used known gene annotations to assign strand, it is likely that some reads originating from un-annotated genes were assigned to a known gene located on the opposite strand. In this case, A-to-G editing events would be identified as T-to-C mismatches, which make up ~8% of the RDD sites and represent the second largest type of nucleotide conversions. Similarly, G-to-A modifications (~6% of sites) could potentially reflect C-to-T editing due to reads assigned to the opposite strand. Together, the four types of modifications (A-to-G, C-to-T, T-to-C, G-to-A) constitute ~92% of all detected RDDs.

However, we cannot exclude the possibility that a few of the non-A-to-G/C-to-T mismatches are indeed SNPs that were not captured in the genotyping data. About 60% of the non-A-to-G and non-C-to-T RDDs were identified in only 1 mouse strain, while only 16% were identified in more than 5 strains. As a comparison, ~44% of A-to-G editing sites were identified in only 1 and ~26% in more than 5 mouse strains.

Importantly, since many of the non-canonical editing sites are identified in only one strain, the ratio of A-to-G editing sites per strain is much larger than the overall mean ratio. Per strain it ranges between 76-87% and is on average 81%.

Generally, the fraction of A-to-G editing sites in mouse is observed to be lower than in humans (mouse: 52-56% Lagarrigue et al., Genetics 2013; mouse: ~60% Gu et al., PLoS One 2012; mouse 96% vs. human 95-99% Porath et al., Nat Commun 2014; mouse 15% vs. human 25% Eisenberg et al., Trends Genet 2005), which may partly be explained by the lower density of Alu elements in the mouse genome (Neeman et al. RNA 2006).

Altogether, we think that the fractions of A-to-G editing sites among all RDDs in our study are as expected and in agreement with previous observations.

*Reviewer #3:*

*In this manuscript, Hasin-Brumshtein and coauthors provide a systems-level analysis of hypothalamic gene expression, its regulation by genetic elements, and its association with metabolic phenotypes. The hypothalamus is composed of diverse and distributed neuronal subpopulations that control multiple aspects of metabolic homeostasis. Here, the authors test the hypothesis that differences in hypothalamic gene expression in a hybrid mouse diversity panel (HMDP) may influence the sensitivity of particular strains to phenotypes associated with diet-induced obesity. The authors provide two specific examples from their data of transcripts (1 annotated and 1 lincRNA) whose expression correlates with a specific metabolic endpoint (Figure 4 and Figure 5). To increase confidence in the analysis, the authors should validate the hypothalamic expression differences between strains for a handful of these transcripts by in situ hybridization.*

*Overall, the lack of functional validation of candidate genes with phenotypic outcomes severely limits this as a resource, and greatly diminishes the impact of this study. Nonetheless, this study does begin offering a detailed mapping of genomic regions underlying differential gene expression and RNA processing events in the hypothalamus, which with additional work could enhance our functional understanding of the hypothalamus. However, in its present form, it is not accessible to the research community working in the hypothalamus.*

*Hasin-Brumshtein et al. characterize the expression of both annotated and novel genes, transcript variants, and lincRNAs from the hypothalamus of mice representing 99 strains from the HMDP. The effort invested in obtaining and analyzing both genomic and proteomic data at this scale is impressive. However, profiling data presented would benefit from a more thorough description of mouse husbandry conditions prior to tissue collection. Were the mice raised on the same chow diet used by Parks et al. (2013) to establish the phenotypic responses of mice in the HMDP to high fat, high sucrose diet? In fact, the authors' description in the Introduction suggests that RNA-Seq was performed on "mice… fed a high fat high sugar diet." If so, it is unclear if the observed expression differences across the mouse strains drive their unique phenotypic responses to dietary challenge or if these differences are a response to such a challenge. Similarly, since RNA-Seq was performed on as few as 1 subject per strain, the authors should elaborate on steps taken to mitigate other environmental factors that might have influenced gene expression in the mice. Why not increase the power of this study by increasing the numbers on the most commonly used strains with the greatest divergence in metabolic outcomes?*

*The authors do acknowledge one major weakness which is the pan-hypothalamic approach, thus reducing their ability to identify genotype/phenotype associations with transcripts (Discussion, sixth paragraph). However, the statement at the end of that paragraph: "transcription… was shown to be largely shared among tissues and cell types" is confusing. If this is true, would not the small differences in expression between subpopulations of neurons be the most important in determining their functional specificity and in turn contribute to phenotypic differences? Such small differences are the least likely to be identified by analyzing the whole hypothalamus. Alternatively, the differences could arise from altered synaptic connectivity, which would also be difficult to resolve by expression profiling. This is an important point that is never discussed.*

*In addition, given that the hypothalamus is a critical region for many important sex-dependent metabolic and behavioral outcomes due to signaling from sexually dimorphic subsets of neurons, one wants to know how these data compare between male and females.*

We thank the reviewer for thoughtful and thorough assessment of our manuscript. This reviewer raised several important issues which we would like to address:

1) We respectfully want clarify what seems to be a conceptual difference between our and reviewer’s view of this work. Particularly, the reviewer seems to assume that the main focus of our effort should be to identify novel transcripts the expression of which is related to the metabolic phenotypes characterized in the HMDP. However, in our view, the main goal of the manuscript is to provide one of a kind resource for researchers that focus on molecular studies of the hypothalamus, as well as system genetics of metabolism. Our studies provide comprehensive characterization of the hypothalamic transcriptome and examples of how this resource can facilitate testing of existing hypothesis, or generate new ones. The study is aimed at exploratory analysis of hypothalamic transcriptome in the mouse – and we hope that it can provide the basis for forming the type of hypothesis the reviewer wishes us to test. We sincerely regret that it was not made clear enough in our manuscript, and have revised the introduction and discussion to reflect this point.

A major critique from reviewer 3 focuses on a lack of functional validation of our presented examples. We completely agree with the reviewer that an association of gene expression with a particular phenotype does not necessarily imply that those changes play any causative role in the development of the phenotype. Also as the reviewer points out, these expression changes could be, or are even likely to be, reactive. This is a general limitation of the many studies that associate gene expression with phenotypes, and is not unique to this work. Nevertheless, previous studies in humans and mice show that identifying expression changes which correlate with a phenotype can often lead to molecular insights and novel hypothesis that can then be tested in follow up experiments. The specific cases of gene-phenotype associations presented in our manuscript were intended as examples of the utility of our data, not as definitive evidence for novel genes regulating these two phenotypes. As such, we believe that the experimental confirmation of these associations is beyond the scope of this study and should be addressed separately. Importantly, since experimental validation is often time consuming and expensive, we suggest the use of multiple lines of evidence when choosing gene targets for experimental confirmation, rather than relying on the association of gene expression alone. To augment the usability of our results, we have added a section discussing the regulation and association of hypothalamic genes already experimentally implicated in metabolic phenotypes by previous studies in mice, or that are markers of particular processes. We identify the regulatory pathways of some of these genes, and show associations to traits that reflect known biology as well as potentially novel aspects.

2) The reviewer raises an important question of tissue heterogeneity in the hypothalamus. The hypothalamus contains many distinct nuclei, that are central to regulation of different phenotypes. Indeed, as stated in our Discussion, using the entire hypothalamus for gene expression analysis is one of the major limitations of our study and is dictated by the physical impracticality of dissecting specific nuclei or neurons in hundreds of mice. Yet, we recognize that this is an important limitation, and to better assess how heterogeneity is reflected in our data, we added a section discussing the expression of hypothalamic cell markers, and their relative abundance across strains. We show that well established markers of hypothalamic neurons (such as Agrp, Sf1, or Pomc genes), as well as glial markers are highly expressed in our data and the variation and genetic regulation of these genes is captured. We further show that despite using whole hypothalamus, we are able to recapitulate the expected relations between hypothalamic transcripts and metabolic traits.

3) Fourth, the reviewer raises a question about whether using only one mouse per strain in 2 of the 99 strains (AxB-12/PgnJ and I/LnJ), is likely to affect the accuracy of our results. We believe that having 2 strains with only one animal out of 99 is highly unlikely to bias our results. However, to address this concern, we used a subset of transcripts to remap eQTLs using only one expression value per strain. This downsampling did not significantly alter our eQTL results. The reviewer also suggests adding additional mice from specific phenotypically diverse inbred strains to enhance the power of the study. We appreciate this suggestion, however our analysis shows that inter-strain variability of gene expression is considerably larger than intra-strain variability. Therefore, increasing the number of individual mice per strain is not likely to increase the power of genetic mapping, as it would not increase the genetic diversity of our sample.

4) The reviewer asked for more details on mouse husbandry conditions, and the relation between our sample and mice used in the Parks et al., 2013 paper. We are grateful for this comment. The mice used in our study are a subset of mice used in Parks et al. study, although in our work we present analysis of a different tissue (hypothalamus). Therefore, the husbandry conditions, the diet and the obtained phenotypes are exactly as were described in Parks et al. 2013 study. To avoid confusion, we have repeated this point in the Materials and methods section, and also made it more clear in the Introduction.

[Editors’ note: the author responses to the re-review follow.]

*Thank you for resubmitting your work entitled "Hypothalamic transcriptomes of 99 mouse strains reveal trans eQTL hotspots, splicing QTLs and novel non-coding genes" for further consideration at eLife. We were receptive to the basis of your appeal, and the revised article has been favorably evaluated by Mark McCarthy (Senior editor), the original Reviewing editor, and by the more critical of the previous sets of reviewers.*

*All of us are positive about your revision. However, there are a couple of issues that remain to be addressed before acceptance, as outlined below:*

*1) The caveat the authors have added to the text is acceptable but the authors should look at Gencode M4 and see if there are overlaps with their genes. This annotation has been available for 2.5 years and is the most current mouse annotation file. M2 dates back to 2013. The rationale that annotation changes too rapidly ("month-to-month") is not supported by the Gencode release dates and since the novelty of these transcripts is a major feature, it should be put in line with the latest (2.5y old) annotation. One could simply overlap novel locations with bed files of latest annotations at a minimum.*

We agree with the reviewer that M2 is quite outdated, and as suggested compared the novelty of our data to the latest released annotation (GENECODE M10, released January 2016), and find that 50% of our novel genes are novel. Please find the relevant additions in the second paragraph of the subsection “Hypothalamic gene expression and proteomic data reveals multiple new isoforms and novel genes”.

2) Another issue that remains is the trans-eQTL results as any SNP correlated with a batch effect can appear as a trans-eQTL hotspot. The trans-eqtl on chr15 relevant to ion transport might be reflecting circadian rhythm. http://www.ncbi.nlm.nih.gov/pmc/articles/PMC2819050/

*The authors should explore this issue in more detail or properly caveat it too.*

We thank the reviewers for these points and agree that we are not able to exclude completely the possibility of unknown batch effects, especially if those would arise from random environmental rather than technical aspects and thus not recorded. However, in our opinion, such factors are unlikely to cause batch effects in a tissue specific manner – thus we revisited the liver and adipose eQTL mapping data, from Parks BW 2013, that used the same mice. We do not observe these trans eQTL hotspots in either adipose or liver data. Nonetheless, we now discuss the possibility of this caveat at length in the fourth paragraph of the Discussion section. We also completely agree that the trans effects acting on ion transport, do not necessarily directly cause the associated changes in metabolic profile. These associations are complex, and can be mediated by variety of physiological processes, and indeed circadian rhythm plays a major role in metabolic homeostasis. We thank the reviewer for this comment, and clarified this point in the Discussion (third paragraph).